# Simplifying Neural Network Training Under Class Imbalance

**Ravid Shwartz-Ziv**[*]
New York University
ravid.shwartz.ziv@nyu.edu

**Micah Goldblum**[*]
New York University
goldblum@nyu.edu

**Yucen Lily Li**
New York University
yucenli@nyu.edu

**C. Bayan Bruss**
Capital One
bayan.bruss@capitalone.com

**Andrew Gordon Wilson**
New York University
andrewgw@cims.nyu.edu

## Abstract

Real-world datasets are often highly class-imbalanced, which can adversely impact the performance of deep learning models. The majority of research on training neural networks under class imbalance has focused on specialized loss functions, sampling techniques, or two-stage training procedures. Notably, we demonstrate that simply tuning existing components of standard deep learning pipelines, such as the batch size, data augmentation, optimizer, and label smoothing, can achieve state-of-the-art performance without any such specialized class imbalance methods. We also provide key prescriptions and considerations for training under class imbalance, and an understanding of why imbalance methods succeed or fail.

## 1 Introduction

Only a minuscule proportion of credit card transactions are fraudulent, and most cancer screenings come back negative. In reality, some events are common while others are exceedingly rare. As a result, machine learning systems, often developed in class-balanced settings [e.g., 48, 46, 16], are routinely trained and deployed on class-imbalanced data where relatively few samples are associated with certain *minority classes*, while *majority classes* dominate the datasets. Class-imbalanced training data can negatively impact performance. Consequently, a wide body of literature focuses on specially tailored loss functions and sampling methods for counteracting the negative effects of imbalance [10, 14, 25, 27, 50, 61]. In striking contrast to such approaches, we instead show that simply tuning existing components of standard neural network training routines can achieve state-of-the-art performance on class-imbalanced image and tabular benchmarks at little implementation overhead and without requiring any specialized loss functions or samplers designed specifically for imbalance. Like Wightman et al. [69], who found that modern training routines allow ResNets to achieve performance competitive with that of later architectures, we show that modern training techniques cause the benefits of specialized class-imbalance methods to nearly vanish.

Moreover, our carefully tuned training routine can be combined with existing class imbalance methods for additional performance boosts. Conducting evaluations on real-world datasets, we find that existing methods, which performed well on web-scraped natural image benchmarks on which they were designed, underperform in the real-world setting, whereas our approach is robust.

Our investigation provides key prescriptions and considerations for training under class imbalance:

---

[*]Authors contributed equally.

37th Conference on Neural Information Processing Systems (NeurIPS 2023).

1. The impact of batch size on performance is much more pronounced in class-imbalanced settings, where small batch sizes shine.

2. Data augmentations have an amplified impact on performance under class imbalance, especially on minority-class accuracy. The augmentation strategies in our experiments which achieve the best performance on class-balanced benchmarks yield inferior performance on imbalanced problems.

3. Large architectures, which do not overfit on class-balanced training sets, strongly overfit on imbalanced training sets of the same size. Moreover, newer architectures which work well on class-balanced benchmarks do not always perform well under class imbalance.

4. Adding a self-supervised loss during training can improve feature representations, leading to performance boosts on class-imbalanced problems.

5. A small modification of Sharpness-Aware Minimization (SAM) [19] pulls decision boundaries away from minority samples and significantly improves minority-group accuracy.

6. Label smoothing [57], especially on minority class examples, helps prevent overfitting.

To understand why exactly such training routine improvements confer significant benefits, we investigate the role of overfitting in class-imbalanced training. Our analysis shows that naive training routines overfit on minority samples, causing neural collapse [60], whereby features extracted from the penultimate layer concentrate around their class-mean. Combining this analysis with decision boundary visualizations, we demonstrate that unsuccessful methods for class-imbalanced training overfit strongly, whereas successful methods regularize.

## 2 Related Work

A long line of research has been conducted on class-imbalanced classification. There are several archetypal approaches specially designed to address imbalance:

**Resampling the data.** In early ensemble learning studies, boosting and bagging algorithms were adjusted to take account of imbalanced data by resampling. Traditionally, resampling involves oversampling minority class samples by simply copying them [25, 10, 27], or undersampling majority classes by removing samples [17, 32, 3, 8], so that minority and majority class samples appear equally frequently in the training process.

**Loss reweighting.** Reweighting methods assign different weights to majority and minority class loss functions, increasing the influence of minority samples which would otherwise play little role in the loss function [14, 34]. For instance, one may scale the loss by inverse class frequency [28] or reweight it using the effective number of samples [14]. As an alternative approach, one may focus on hard examples by down-weighing the loss of well-classified examples [50] or dynamically rescaling the cross-entropy loss based on the difficulty of classifying a sample [61]. Bertsimas et al. [6] encourage larger margins for rare classes, while Goh and Sim [21] learn robust features for minority classes using class-uncertainty information which approximates Bayesian methods.

**Two-stage fine-tuning and meta-learning.** Two-stage methods separate the training process into representation learning and classifier learning [54, 59, 38, 4]. In the first stage, the data is unmodified, and no resampling or reweighting is used to train good representations. In the second stage, the classifier is balanced by freezing the backbone and fine-tuning the last layers with resampling or by learning to debias the class confidences. These methods assume that the bias towards majority classes exists only in the classifier layer or that tweaking the classifier layer can correct the underlying biases.

Several works have also inspected representations learned under class imbalance. Kang et al. [38] find that representations learned on class-imbalanced training data via supervised learning perform better when the linear head is fine-tuned on balanced samples. Yang and Xu [71] instead examine the effect of self- and semi-supervised training on imbalanced data and conclude that imbalanced labels are significantly more useful when accompanied by auxiliary data for semi-supervised learning. Kotar et al. [44], Yang and Xu [71] make the observation that self-supervised pre-training is insensitive to imbalance in the upstream training data. These works study SSL pre-training for the purpose of transfer learning, sometimes using linear probes to evaluate the quality of representations. Inspired by their observations, we find that the addition of an SSL loss function on the same class-imbalanced dataset, even when no upstream data is available, can significantly improve generalization.

In summary, existing works propose countless approaches to address class imbalance during training. In contrast, we show that strong performance can be achieved on class-imbalanced datasets simply by tuning the components of standard neural networks training routines, without specialized loss functions or sampling methods designed specifically for imbalance. Our tuned routines require little to no additional implementation compared to standard neural network training pipelines and can be combined with existing specialized approaches for class imbalance. We additionally provide novel prescriptions and considerations for training under class imbalance, as well as an understanding of how regularization can contribute to the success of training under imbalance.

# 3 Optimizing Training Routines for Imbalanced Data

While previous research on training class imbalance mainly concentrated on developing new loss functions and sampling methods tailored to imbalanced data, little attention has been paid to how the components of traditional training routines interact with such data.

In this section, we delve into various methods that can be optimized or modified specifically for imbalanced training scenarios. For clarity and depth of discussion, we divide these methods into two distinct groups, each discussed in its own subsection.

Subsection 3.1 deals with what we identify as the "fundamental building blocks" of conventional balanced training. This includes elements like batch size, data augmentation, pre-training, model architecture, and optimizers. We scrutinize the effects of altering these elements' hyperparameters, emphasizing their influence on the performance of models under imbalanced training. Our discussion is fortified by experiments conducted on both vision datasets (CIFAR-10, CIFAR-100, CINIC-10 and Tiny-ImageNet) and tabular datasets across different network architectures.

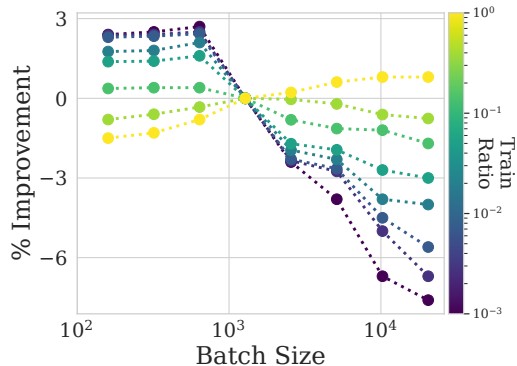

**Figure 1: Imbalanced data prefers small batch sizes.** We plot the percent improvement in accuracy over the baseline batch size of 128 for different train ratios as a function of batch size. Positive values indicate higher accuracy than the baseline. Balanced training sets yield flatter lines, indicating insensitivity to batch size. Experiments conducted with ResNet-50 on CIFAR-100.

Subsection 3.2 introduces a set of optimization methods—Joint-SSL, SAM, and label smoothing—that we have reformed to cater more appropriately to imbalanced training. These methods, initially designed for balanced datasets, are subjected to modifications making them more amenable to imbalanced scenarios. The effectiveness of these methods, including comparisons to other state-of-the-art techniques across various domains and datasets (vision and tabular), is thoroughly evaluated in Section 3.1.1.

This structure allows a separate discourse on the fundamental aspects of deep learning training and the particular adjustments needed for imbalanced training. Our aim is to provide a comprehensive overview of both traditional and innovative techniques, giving readers a broad spectrum of strategies for tackling imbalanced data.

## 3.1 Tuning the Building Blocks of Neural Network Training

The success of deep learning models hinges on the precise orchestration of their training routine. The success of models is especially sensitive to this orchestration under imbalanced training data. We now explore the building blocks of neural network training routines and their importance in optimizing models for imbalanced data. This study equips researchers and practitioners with practical strategies to improve the performance of their models under class imbalance. In this subsection, we focus on batch size, data augmentation, pre-training, model architecture, and optimizers.

### 3.1.1 Experimental Setup

**Datasets.** To conduct our investigation, we leverage three benchmark image datasets: CIFAR-10 [46], CIFAR-100, and CINIC-10 [15], along with three tabular datasets: Otto Group Product Classification [37], Covertype [7], and Adult datasets [43]. For naturally balanced datasets, we adopt the imbalanced setup proposed by Liu et al. [54], which employs an exponential distribution to imbalance classes, closely mirroring real-world long-tailed class distributions. The **Class-imbalance ratio** ($r$) represents the ratio of samples in the rarest to the most frequent class. A dataset with $r = 1$ is fully balanced, while $r = 0.1$ indicates that the majority class has ten times more samples than the minority class. We investigate varying imbalance ratios in both the training and test sets.

**Models.** For image datasets, we utilize ResNets [29] and WideResNet [73] with different depths (8, 32, 50, and 152). In addition, to evaluating the role of architecture in imbalanced training, we also employ DenseNet [35], MobileNetV2 [62], Inception v3 [67], EfficientNet [68], and VGG [64]. Tabular datasets are processed using XGBoost [11], SVM, and MLP. We follow the supervised pre-training protocol of Kang et al. [39], while for self-supervised pre-training, we employ SimCLR [12] and VICReg [5], where the fine-tuning is as described in Kotar et al. [44]. Further details can be found in Appendix B.

**Metrics.** In addition to overall test accuracy, we provide minority and majority class accuracy (representing the 20% of classes with the smallest and largest number of samples) in the appendix, if not in the main body of the paper. We conduct five runs with different seeds for each evaluation in our experiments and report the mean along with one standard error.

### 3.1.2 Results

**Batch size.** Studies conducted on balanced training data suggest that small batch sizes may exhibit superior convergence behavior, while large batch sizes can reach optimal minima unattainable by smaller sizes [41]. In class-imbalanced settings, one intuition is that larger batch sizes may be necessary to obtain enough samples from the minority class and counteract forgetting. On the other hand, large batches can also increase the risk of overfitting.

To examine the influence of batch size in class-imbalanced contexts, we train networks with varying batch sizes across multiple training ratios. In Figure 1, we plot the percent improvement in accuracy over the baseline (which is set as best batch size 128 as a function of batch size for several training ratios. Positive values represent higher accuracy compared to the baseline, whereas negative values denote lower accuracy.

Our analysis reveals that batch size has a much greater impact in highly imbalanced settings and very little impact in balanced settings. Notably, data with a high degree of class imbalance tends to benefit from smaller batch sizes, even though small batches often do not contain minority samples, possibly due to the regularization effects that help mitigate overfitting to the majority classes. See Appendix A.1 for additional details and experiments.

**Data augmentation.** Data augmentation is a feature of virtually all modern image classification pipelines. We now investigate the impacts of various augmentation policies across varying levels of class imbalance. Our experiments show that the effects of data augmentation are greatly amplified on imbalanced data, especially for minority classes (Figure 2 - left). This finding suggests that augmentation serves as a regularizer, supporting recent studies on the role of data augmentation in preventing overfitting during class-balanced training [20]. Moreover, we find that the optimal augmentation policy can depend on the level of imbalance.

We assess our findings using a variety of augmentation methods including horizontal flips, random crops, AugMix [31], TrivialAugmentWide [58], AutoAugment [13], Mixup, CutOut, and Random Erasing.

To identify the most potent data augmentation strategy, we gauge the improvement in accuracy, represented as the percentage increase compared to training without augmentation, in Appendix A.2. While TrivialAugment outperforms other methods on balanced training data, AutoAugment emerges as the most effective for imbalanced data

Furthermore, we examine the sensitivity of performance to the specific augmentation method used. By assessing the variance in performance across different augmentations for balanced and imbalanced

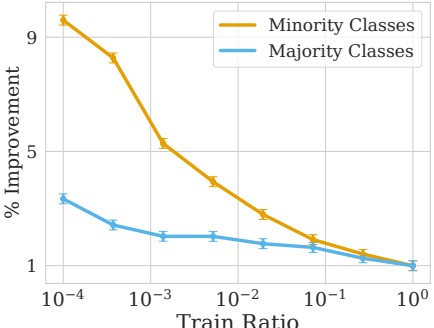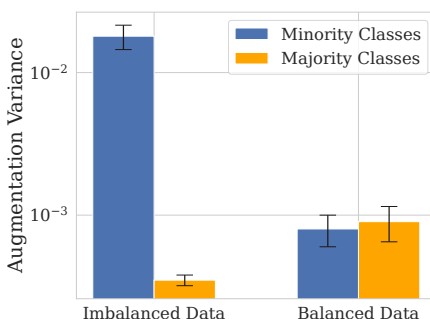

Figure 2: **Left: Augmentations yield far bigger improvements on minority classes.** We compare the percent improvement in test accuracy of TrivialAugment compared to training without any augmentation as a function of the training ratio. **Right: The type of augmentation matters more on imbalanced data.** *Augmentation variance* measures the variance in percent improvement over training without augmentation across different augmentation types. Variance across augmentation types on imbalanced data is much greater for minority classes, indicating the importance of choosing an appropriate augmentation policy. Experiments conducted with ResNet-50 on CIFAR-100. Error bars represent one standard error over 5 trials.

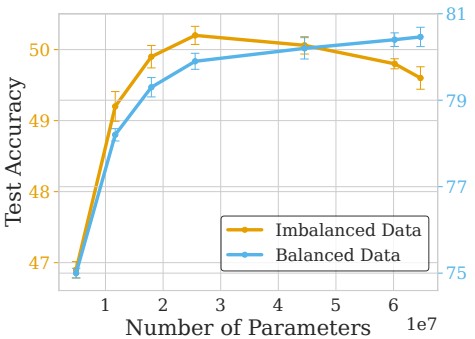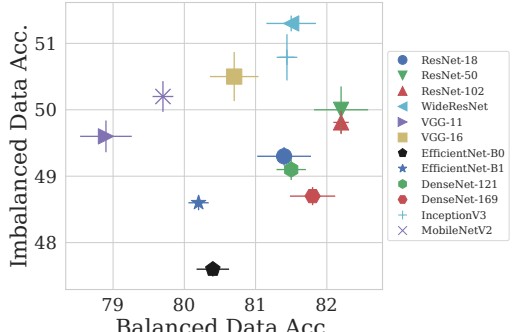

Figure 3: **Left: Deeper architectures overfit on class-imbalanced data.** While deeper ResNet models perform better on balanced data, they can overfit and underperform on imbalanced data. **Right: Performance on balanced and imbalanced datasets is virtually uncorrelated across a wide variety of architectures** (Pearson correlation coefficient 0.14). Experiments conducted on CIFAR-100 with an imbalanced train ratio of 0.001. Error bars represent one standard error over 5 trials.

situations (see Figure 2), we find that minority class performance is particularly sensitive to the chosen augmentation policy See Appendix A.2 for additional details and experiments.

**Model architecture.** While larger networks often enhance performance on class-balanced datasets without overfitting, their efficacy on imbalanced data remains unexplored. To probe this behavior, we train ResNets of various sizes on both balanced and imbalanced data with a training ratio of 0.01. We plot the test accuracy on minority classes as a function of the network's size in Figure 3 (left). While the network's performance on balanced data monotonically improves with increasing size, its performance on imbalanced data peaks at a size with 20 million parameters, and declines thereafter. This dip in performance suggests that, unlike their behavior on balanced data, larger networks may be susceptible to overfitting minority classes in the face of severe class imbalance. We then plot the test accuracy of a wide variety of architectures on the balanced and imbalanced CIFAR-100 variants in Figure 3 (right), and find that accuracies in these two settings are virtually uncorrelated! Computer vision architectures have largely been developed on well-known class-balanced benchmarks like ImageNet [16]. These developments may not generalize to imbalanced settings which are common in the real world. See Appendix A.3 for additional details and experiments.

**Pre-training.** Fine-tuning models pre-trained on expansive upstream datasets can markedly enhance performance across a variety of domains by equipping the model with an informative initial parameter

vector learned from pre-training data [12, 65]. Self-Supervised Learning (SSL) has emerged as a highly effective strategy for representation learning in fields such as computer vision, natural language processing (NLP), and tabular data. Networks pre-trained using SSL often produce more transferable representations than those pre-trained through supervised learning. Furthermore, self-supervised pre-training strategies for transfer learning display greater robustness to upstream imbalance compared to supervised pre-training [51]. We observe in our experiments that pre-trained backbones provide considerably greater benefits under severe downstream class imbalance than under balanced downstream training sets. The ability for SSL pre-training to mitigate overfitting could therefore be particularly valuable in the class-imbalanced setting.

To gauge the effectiveness of pre-training, we fine-tune various pre-trained models on downstream datasets with a range of class imbalance ratios. We make use of pre-trained ResNet-50 weights learned on ImageNet-1K [16], ImageNet-21K, and two self-supervised learning methods, SimCLR and VICReg. Figure 4 depicts the relative improvement in test accuracy compared to training from random initialization across different pre-training models. A positive value signifies a performance improvement. All pre-training methods notably outperform random initialization. However, we observe a considerably larger improvement under class imbalanced scenarios, where models pre-trained on larger datasets yield greater boosts in accuracy. Moreover, SSL methods conducted on ImageNet-1K surpass supervised pre-training on the same upstream training set. See Appendix A.4 for additional details and experiments.

## 3.2 Improved Optimization Methods for Class Imbalance

While the fundamental building blocks above provide a strong foundation for training performant models, we can also customize modern optimization techniques specifically for imbalanced training and achieve further gains. This subsection showcases adapted variants of SSL, SAM, and label smoothing.

**Self-Supervision.** Self-Supervised Learning (SSL) has received substantial attention in representation learning, particularly in computer vision, NLP, and tabular data [12, 40, 65, 49], especially for training on massive volumes of unlabeled data. Networks pre-trained via SSL often showcase more transferable representations compared to those pre-trained through supervised methods [24]. Moreover, self-supervised pre-training strategies for transfer learning exhibit more robustness to upstream imbalance compared to supervised pre-training [51]. Despite these advantages, the availability of massive pre-training datasets can often be a limiting factor in many use cases.

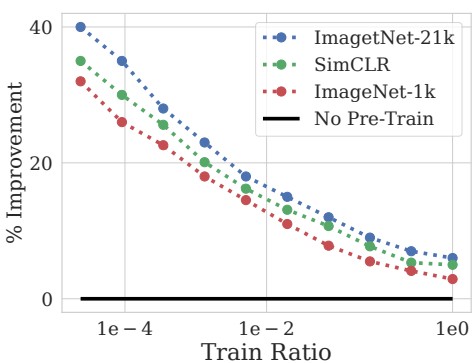

**Figure 4: Pre-training is more impactful on imbalanced downstream data.** *% Improvement* refers to the improvement in test accuracy compared to training from random initialization. The benefits of SSL over supervised pre-training are also amplified under class imbalance. Experiments conducted with ResNet-50 on CIFAR-100.

Traditionally, pre-training involves a two-step process: learning on an upstream task, followed by fine-tuning on a downstream task. In our approach, we instead integrate supervised learning with an additional self-supervised loss function during from-scratch training, bypassing the need for pre-training. The simple integration of an SSL loss function, which does not depend on class-imbalanced labels and is insensitive to imbalance, results in improved feature representations and better generalization. We refer to this combined training procedure as Joint-SSL. It is important to note that our method differs from those used in Kotar et al. [44], Yang and Xu [71], Liu et al. [51], which investigate SSL pre-training on larger datasets for transfer learning, rather than directly training with an SSL objective on downstream data. See Appendix A.5 for additional details.

**Sharpness-Aware Minimization (SAM).** SAM [19] is an optimization technique for finding "flat" minima of the loss function, often improving generalization. The technique involves taking an inner ascent step followed by a descent step to find parameters that minimize the increase in loss from the ascent step. Huang et al. [36] shows that flat minima correspond to wide-margin decision boundaries.

We thus adapt SAM for class-imbalanced cases by increasing the flatness especially for minority class loss terms. To do so, we increase the ascent step size in SAM's inner loop for minority classes, denoting this method SAM-Asymmetric (SAM-A). Plotting the decision boundaries of a small multi-layer perceptron on a toy 2D dataset (Appendix Figure 18), we observe that classifiers naturally form small margins surrounding minority class samples. SAM-A expands these margins, preventing the model from overfitting to the minority samples. See Appendix A.6 for additional details.

**Label smoothing.** Conventionally, classifiers are trained with hard targets, minimizing the cross-entropy between true targets $y_k$ and network outputs $p_k$ as in $H(y; p) = \sum_{k=1}^{K} -y_k \log(p_k)$, with $y_k$ equal to 1 for the correct class and 0 otherwise. Label smoothing uses a smoothing parameter, $\epsilon$ to instead minimize the cross-entropy between smoothed targets $y_{LS_k}$ and network outputs $p_k$, where $y_{LS_k} = y_k(1 - \epsilon) + \epsilon/K$ [57].

We adapt label smoothing for the class-imbalanced setting by applying more smoothing to minority-class examples than to majority-class examples. This procedure prevents overfitting on minority samples. See Appendix A.7 for additional details and experiments.

**Dataset curation.** Common intuition dictates that training on data that is more balanced than the testing distribution can improve representation learning by preventing overfitting to minority samples [25, 10, 27]. In Appendix A.8, we find that this intuition may be misguided both for neural networks and gradient-boosted decision trees, especially on large datasets, and that curating additional samples may in fact be destructive if the proper dataset balance is not maintained.

# 4 Benchmarking Training Routines under Class Imbalance

In the preceding sections, we examined various building blocks of balanced training routines and presented modifications of optimization methods—label smoothing, Sharpness-Aware Minimization, and self-supervision—tailored specifically for imbalanced scenarios. In this section, we compare the performance of models trained using our modified methods to those trained using existing state-of-the-art methods for handling class imbalance. To ensure a fair and unbiased comparison, all methods are trained using the same refined training routines. Our comparison provides evidence of the effectiveness of our proposed methods when coupled with our optimized routines.

## 4.1 Vision Datasets

### 4.1.1 Experimental Setup

**Datasets.** We perform experiments on six image datasets, including natural image, medical, and remote sensing datasets: CIFAR-10 [46], CIFAR-100, CINIC-10 [15], Tiny-ImageNet[16], SIIM-ISIC Melanoma [26], APTOS 2019 Blindness [1] and EuroSAT [30].

**Baseline Methods.** We compare to the following comprehensive range of baselines: (a) **Empirical Risk Minimization (ERM)** involves training on the cross-entropy loss without any re-balancing. (b) **Resampling** balances the objective by adjusting the sampling probability for each sample. (c) **Synthetic Minority Over-sampling Technique (SMOTE)** [10] is a re-balancing variant that involves oversampling minority classes using data augmentation. (d) **Reweighting** [33] simulates balance by assigning different weights to the majority and minority classes. (e) **Deferred Reweighting (DRW)** involves deferring the resampling and reweighting until a later stage in the training process. (f) **Focal Loss (Focal)** [50] upweights the objective for difficult examples, thereby focusing more on the minority classes. (g) **Label-Distribution-Aware Margin (LDAM-DRW)** [9] trains the classifier to impose a larger margin on minority classes. (h) **M2m** [42] translates samples from majority to minority classes. Lastly, (i) **MiSLAS** [75] is a two-stage training method that combines mixup [74] with label smoothing. We also combine our techniques with previous state-of-the-art Major-to-minor Translation (M2m) [42] and observe it improves performance over M2m alone.

**Evaluation.** We follow the evaluation protocol used in [54, 42], training models on class-imbalanced training sets and evaluating them on balanced test sets. We evaluate on four benchmark datasets for imbalanced classification: CIFAR-10, CIFAR-100 [46], CINIC-10 [15] and Tiny-ImagNet [47] with training ratios of 0.01, 0.02, and 0.1. Additionally, we use three real-world datasets, namely APTOS 2019 Blindness Detection [1], SIIM-ISIC Melanoma Classification [26], and EuroSAT [30].

**Table 1: Our training routines exceed previous SOTA or improve existing methods when combined.** Accuracy on various datasets using different methods. Error bars correspond to one standard error over 5 trials.

| Method | CIFAR-10 0.01 | CIFAR-10 0.02 | CIFAR-100 0.01 | CIFAR-100 0.02 | CINIC-10 0.01 | CINIC-10 0.02 | Tiny-ImageNet (0.1) SwinV2 | Tiny-ImageNet (0.1) ConvNeXt |
|---|---|---|---|---|---|---|---|---|
| ERM | $84.9 \pm 0.2$ | $84.2 \pm 0.3$ | $47.7 \pm 0.5$ | $52.5 \pm 0.4$ | $78.6 \pm 0.2$ | $82.3 \pm 0.3$ | $53.4 \pm 0.2$ | $53.1 \pm 0.3$ |
| Reweighting | $81.8 \pm 0.3$ | $79.6 \pm 0.4$ | $42.1 \pm 0.4$ | $47.8 \pm 0.3$ | $71.4 \pm 0.4$ | $74.9 \pm 0.4$ | $52.8 \pm 0.4$ | $52.3 \pm 0.1$ |
| Resampling | $82.1 \pm 0.3$ | $79.0 \pm 0.2$ | $42.8 \pm 0.3$ | $48.2 \pm 0.4$ | $72.1 \pm 0.5$ | $75.6 \pm 0.4$ | $52.5 \pm 0.3$ | $52.1 \pm 0.2$ |
| Focal Loss | $83.2 \pm 0.5$ | $84.4 \pm 0.4$ | $47.1 \pm 0.5$ | $52.9 \pm 0.5$ | $74.9 \pm 0.4$ | $78.4 \pm 0.3$ | $53.5 \pm 0.1$ | $53.1 \pm 0.4$ |
| LDAM-DRW | $84.9 \pm 0.3$ | $82.9 \pm 0.4$ | $47.2 \pm 0.4$ | $53.3 \pm 0.3$ | $76.0 \pm 0.3$ | $79.7 \pm 0.4$ | $54.2 \pm 0.2$ | $53.4 \pm 0.3$ |
| M2m | $84.5 \pm 0.2$ | $84.3 \pm 0.2$ | $47.1 \pm 0.3$ | $52.1 \pm 0.3$ | $78.3 \pm 0.5$ | $81.3 \pm 0.5$ | $54.3 \pm 0.4$ | $53.9 \pm 0.2$ |
| MiSLAS | $85.1 \pm 0.3$ | $84.3 \pm 0.3$ | $47.3 \pm 0.3$ | $53.1 \pm 0.5$ | $78.1 \pm 0.2$ | $81.8 \pm 0.3$ | $54.1 \pm 0.3$ | $53.4 \pm 0.1$ |
| SAM-A | $85.4 \pm 0.3$ | $83.3 \pm 0.3$ | $47.1 \pm 0.4$ | $52.7 \pm 0.2$ | $77.3 \pm 0.2$ | $80.9 \pm 0.2$ | $54.7 \pm 0.4$ | $53.9 \pm 0.2$ |
| Joint-SSL | $85.2 \pm 0.2$ | $84.1 \pm 0.2$ | $47.0 \pm 0.2$ | $51.7 \pm 0.5$ | $77.5 \pm 0.3$ | $81.3 \pm 0.3$ | $54.3 \pm 0.2$ | $53.7 \pm 0.3$ |
| Joint-SSL + SAM-A + Smoothing | $85.9 \pm 0.2$ | $84.7 \pm 0.3$ | $48.0 \pm 0.3$ | $52.6 \pm 0.3$ | $78.1 \pm 0.6$ | $\mathbf{82.3 \pm 0.4}$ | $54.8 \pm 0.1$ | $54.1 \pm 0.4$ |
| Joint-SSL + SAM-A + M2m | $\mathbf{86.0 \pm 0.5}$ | $\mathbf{85.0 \pm 0.4}$ | $\mathbf{48.9 \pm 0.2}$ | $\mathbf{53.8 \pm 0.2}$ | $\mathbf{79.2 \pm 0.4}$ | $82.1 \pm 0.3$ | $\mathbf{55.0 \pm 0.2}$ | $\mathbf{54.3 \pm 0.3}$ |

**Training procedure.** Our training protocol for all methods incorporates TrivialAugment [58] + CutMix [72] as an augmentation policy, along with label smoothing and an exponential moving weight average with small batch size on image classification. We use WideResNet-28×10 [73] for CIFAR-10, CIFAR-100, and CINIC-10 datasets. For Tiny-ImageNet we use both ConvNeXt [53] and Swin transformer v2 [52] and ResNeXt-50-32×4d [70] for the real-world datasets following the comparisons in Fang et al. [18]. We provide further implementation details in Appendix B.

### 4.1.2 Image Classification Benchmarks

We see in Table 1 that our adaptations alone bring the performance of standard ERM training remarkably close to or even exceeding that of state-of-the-art imbalanced training methods. Additionally, the integration of our findings from Section 3 with Joint-SSL and Asymmetric-SAM atop M2m establishes a new state-of-the-art across all benchmark datasets. Even using only our self-supervision and Asymmetric-SAM (without M2m) often surpasses the previous state-of-the-art. Furthermore, the integration of our previous findings from Section 3 with Joint-SSL and Asymmetric-SAM atop M2m establishes new state-of-the-art performance across all benchmark datasets.

Our combined method enhances minority class accuracy while preserving accuracy on majority classes. On the 20 smallest classes of CIFAR-100, our method reduces error by $5.4\% - 6.8\%$ compared to baseline methods. On CINIC-10, our approach enhances the generalization on the two smallest minority classes by $4.6\% - 8.5\%$ compared to baseline methods. Additional results regarding split class accuracy and different backbones can be found in Appendix A.9.

### 4.1.3 Real-World Datasets

Since the majority of research on class-imbalanced training focuses on web-scraped datasets such as ImageNet or CIFAR-10 and CIFAR-100, we now investigate whether such advances are overfit to these datasets or whether they are robust in other settings.

To address this question, we consider the SIIM-ISIC Melanoma, APTOS 2019 Blindness, and EuroSAT datasets. In Table 2, we observe that our highly tuned training routine equipped with Joint-SSL, Asymmetric-SAM, and modified label smoothing delivers equal or superior performance compared to previous state-of-the-art imbalanced training methods on all datasets.

**Correlation between performance on different datasets.** Given the superior performance of

**Table 2: Our routines exceed previous SOTA on real-world datasets.** Error bars corresponds to one standard error over 5 trials using ResNeXt-50-32×4. ↑ denotes the method on the line above.

| Method | SIIM-ISIC Melanoma | APTOS 2019 Blindness | EuroSAT |
|---|---|---|---|
| ERM | $95.1 \pm 0.3$ | $89.1 \pm 0.3$ | $99.0 \pm 0.2$ |
| Reweighting | $94.7 \pm 0.3$ | $88.4 \pm 0.2$ | $98.4 \pm 0.3$ |
| Resampling | $94.9 \pm 0.2$ | $88.7 \pm 0.4$ | $98.7 \pm 0.3$ |
| Focal Loss | $95.0 \pm 0.2$ | $89.2 \pm 0.3$ | $98.9 \pm 0.3$ |
| LDAM-DRW | $95.2 \pm 0.2$ | $89.0 \pm 0.4$ | $99.0 \pm 0.4$ |
| M2m | $95.2 \pm 0.2$ | $89.6 \pm 0.5$ | $\mathbf{99.3 \pm 0.2}$ |
| MiSLAS | $95.4 \pm 0.2$ | $89.4 \pm 0.4$ | $99.2 \pm 0.3$ |
| Joint-SSL | $95.7 \pm 0.3$ | $89.7 \pm 0.5$ | $99.1 \pm 0.2$ |
| ↑ + SAM-A | $96.0 \pm 0.2$ | $\mathbf{90.1 \pm 0.6}$ | $99.1 \pm 0.2$ |
| ↑ + Smoothing | $\mathbf{96.2 \pm 0.2}$ | $\mathbf{90.1 \pm 0.6}$ | $\mathbf{99.3 \pm 0.2}$ |

our approach on these datasets, we now examine the correlation between the performance of various methods on CIFAR-10 (with a training ratio of 0.01) and their performance on real-world datasets. Our findings, illustrated in Table 3, reveals a surprisingly low correlation. This low correlation suggests that a method's successful application to web-scraped datasets like CIFAR-10 does not necessarily translate into equivalently strong performance on real-world datasets. These findings further underscore the need for a more diverse range of datasets in the development and testing of machine learning methods for class imbalance.

## 4.2 Tabular Datasets

Tabular data problems represent a challenging frontier for deep learning research. While recent advances in natural language processing (NLP), vision, and speech recognition have been driven by deep learning models, their efficacy in the tabular domain remains unclear. Notably, there is a debate over the performance of neural networks in comparison to decision tree ensembles like XGBoost [11, 63, 56]. Despite the fact that most tabular datasets inherently exhibit imbalance, there has been limited research addressing the impact of imbalanced data on deep learning in tabular domains.

Table 3: There is only a weak correlation between CIFAR-10 test accuracy and performance on real-world datasets.

| Dataset | Correlation | Slope |
|---|---|---|
| APTOS 2019 Blindness | 0.11 | 0.04 |
| SIIM-ISIC Melanoma | 0.19 | 0.03 |
| EuroSAT | 0.03 | 0.04 |

We thus apply our findings to imbalanced tabular datasets, using a Multilayer Perceptron (MLP) with the improved numerical feature embeddings of Gorishniy et al. [23]. Our approach incorporates SAM-A, modified label smoothing, and SGD with cosine annealing performed and small batch size.

We apply our methods to the following three imbalanced tabular datasets: Otto Group Product Classification [37], Covertype [7], and Adult datasets [43]. We compare our methodology with the following baseline methods: (1) XGBoost, (2) MLP, (3) ResNet, and (4) FT-Transformer, where the last three baselines are employed as in Gorishniy et al. [22]. Our tuning, training, and evaluation protocols are consistent with those in Gorishniy et al. [23]. For full details about the training procedures, see Appendix A.9.1.

We see in Table 6 that our tuned training routine outperforms both XGBoost and recent state-of-the-art neural methods on all datasets, demonstrating the applicability of our findings beyond image classification.

## 5 Regularization and Overfitting in Class-Imbalanced Training

Specialized loss functions and sampling methods for class-imbalanced learning are often designed to mitigate overfitting [42], yet we rarely look under the hood to understand what happens to models trained in this setting. In this section, we quantify and visualize overfitting during class-imbalanced training, and we find that successful methods regularize against this overfitting.

One concern for training under class imbalance might be optimization. Perhaps minority samples are hard to fit since they occur infrequently in batches during training. In Appendix A.10, we verify that empirical risk minimization, without any special interventions, easily fits all training data. This observation indicates that variations in performance among the different methods we compare stem not from their optimization abilities but from their generalization to unseen test samples.

To understand the differences between classifiers learned on imbalanced training data, we visualize their decision boundaries on a 2D toy problem with a multi-layer perceptron. Standard training results in small margins around minority class samples, whereas SAM-A, acting as a regularizer, expands these margins (Figure 18a). For additional examples, see Appendix A.11, where we also use the method introduced by Somepalli et al. [66] to visualize decision boundaries.

To quantify these observations, we examine the Neural Collapse phenomenon [45], which was previously observed in the class-balanced setting. Neural Collapse refers to the tendency of the features in the penultimate layer associated with training samples of the same class to concentrate around their class-means. Our investigation focuses on two metrics:

**Class-Distance Normalized Variance (CDNV)**: This metric evaluates the compactness of features from two unlabeled sample sets, $S_1$ and $S_2$, relative to the distance between their respective feature means. A value trending towards zero indicates optimal clustering.

**Nearest Class-Center Classifier (NCC)**: As training progresses, feature embeddings in the penultimate layer undergoing Neural Collapse become distinguishable. Consequently, the classifier tends to align with the 'nearest class-center classifier'.

We see in Table 4 that the collapse, measured by CDNV and NCC for minority training examples, is significantly worse in standard training without class-imbalance interventions. Furthermore, a correlation exists between the degree of collapse and the performance of the different methods. Specifically, methods that successfully counteract Neural Collapse exhibit superior performance.

We conclude that well-tuned training routines can regularize and prevent overfitting to minority class training samples without specialized loss functions or sampling methods, which is associated with performance improvements on class-imbalanced data. See Appendix A.10 for more details.

**Table 4: Neural Collapse during class-imbalanced training.** Neural collapse (low CDNV and high NCC) corresponds to low test accuracy. Experiments conducted on CIFAR-10.

| Method | CDNV | NCC | Accuracy |
|---|---|---|---|
| ERM | $0.42 \pm 0.02$ | $0.92 \pm 0.02$ | $84.58 \pm 0.40$ |
| Reweighting | $0.38 \pm 0.02$ | $0.94 \pm 0.01$ | $82.62 \pm 0.44$ |
| Resampling | $0.38 \pm 0.03$ | $0.97 \pm 0.01$ | $82.16 \pm 0.43$ |
| LADM-DRW | $0.41 \pm 0.02$ | $0.94 \pm 0.01$ | $83.86 \pm 0.23$ |
| Joint-SSL + SAM-A | $0.43 \pm 0.02$ | $0.90 \pm 0.02$ | $84.80 \pm 0.43$ |

## 6 Discussion

While neural network training practices have been studied extensively on class-balanced benchmarks, real-world problems often involve class imbalance. We have shown that class-imbalanced datasets require carefully tuned batch sizes and smaller architectures to avoid overfitting, as well as specially chosen data augmentation policies, self-supervision, sharpness-aware optimizers, and label smoothing. Whereas previous state-of-the-art works for class-imbalance focused on specialized loss functions or sampling methods, we show that simply tuning standard training routines can significantly improve performance over such ad hoc approaches.

Our findings give rise to several important directions for future work:

- We saw that existing methods designed for web-scraped natural image classification benchmarks do not always provide improvements on other real-world problems. If we are to reliably compare methods for class imbalance, we need a more diverse benchmark suite.

- Since most real-world datasets are class-imbalanced and architectures designed for class-balanced benchmarks like ImageNet are highly suboptimal under class imbalance, perhaps future work should build architectures which are specifically optimized for class imbalance.

- The generalization theory literature explains the tradeoff between fitting training samples and the complexity of learned solutions [55]. Can PAC-Bayes generalization bounds explain the large role regularization plays in successful training under class imbalance?

- Language models perform classification over tokens, but some tokens occur much less frequently than others. How can we apply what we have learned about training under class imbalance to language models?

**Acknowledgements.** This work is supported by NSF CAREER IIS-2145492, NSF I-DISRE 193471, NIH R01DA048764-01A1, NSF IIS-1910266, NSF 1922658 NRT-HDR, Meta Core Data Science, Google AI Research, BigHat Biosciences, Capital One, and an Amazon Research Award.

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

## Appendix Outline

. This appendix is organized as follows:

- In Appendix A, we provide more detailed results on additional datasets for the different components discussed in Section 3. Specifically, in subsections Appendices A.1 to A.7, we present results on batch size, data augmentation, model architectures, pre-training, SSL, Sharpness-Aware Minimization, and label smoothing. These subsections delve into the specific effects and outcomes of each component.

- In Appendix A.8, we examine the relationship between the training and test distributions in imbalanced training. We explore the optimal balance of training data and discuss the potentially destructive impact of collecting additional majority samples.

- In Appendix A.9, we present additional and extended experimental results that compare the methods proposed in our paper with the baseline methods.

- In Appendix A.10, we provide additional experimental results that illustrate how the training process evolves for imbalanced data.

- In Appendix A.11, we include decision boundary visualizations for imbalanced training. Specifically, we demonstrate that Sharpness-Aware Minimization (SAM-A) helps decision regions take up similar volumes, whereas standard training routines tend to shrink-wrap the decision boundaries around minority samples.

- In Appendix B, we provide detailed information on the hyperparameters, datasets, and architectures used in our experiments.

- In Appendix C, we discuss the limitations of our study. This section addresses potential constraints, challenges, and areas for improvement in our research.

- Lastly, in Appendix D, we discuss the broader impact of our work. This section explores the implications, significance, and potential applications of our findings beyond the scope of the immediate study.

## A  Additional Experiments

### A.1  Batch Size

To investigate the impact of batch size in the context of class imbalance, we train networks across various training ratios using different batch sizes. In order to compare the accuracy for each training ratio, we calculate the percentage improvement over the baseline (set as the best batch size of 128). Specifically, if we denote $Acc_b^\rho$ as the accuracy on the imbalanced dataset with training ratio $\rho$ and batch size $b$, we define the adjusted accuracy $new_A cc_b^\rho$ as

$$\bar{Acc}_b^\rho = \frac{Acc_b^\rho - Acc_{128}^\rho}{Acc_b^\rho}. \tag{1}$$

Positive values represent higher accuracy compared to the baseline, while negative values denote lower accuracy. This normalization allows us to examine the relative effect of batch size. As shown in the main text and Figure 5, data with a high degree of class imbalance tends to benefit from smaller batch sizes, despite the fact that small batches often do not contain any minority samples.

### A.2  Data Augmentation

In order to evaluate and compare the effectiveness of various popular augmentation techniques—including horizontal flips, random crops, AugMix [31], TrivialAugmentWide [58], and AutoAugment [13]—we investigate their impact on the accuracy of minority and majority classes across a range of training ratios.

We measure the relative improvement in performance by comparing the accuracy achieved with data augmentation to that achieved without it. We thus plot the percentage improvement as a function of the training ratio in Figure 6.

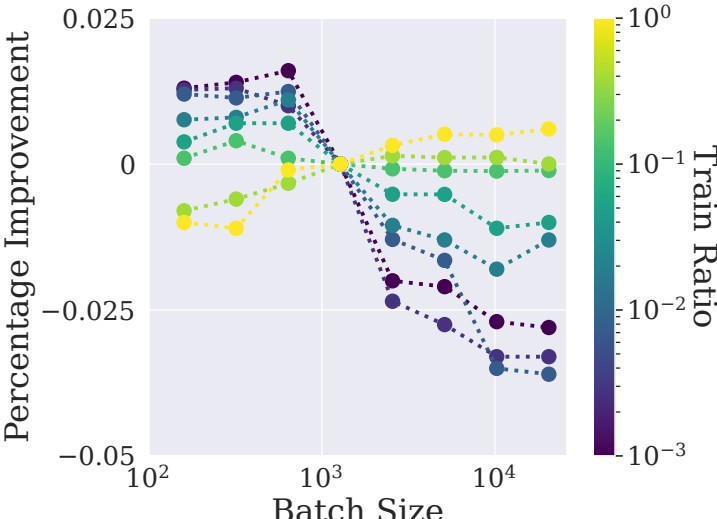

**Figure 5: Batch size matter more for imbalanced data where small batch sizes are best, whereas the curve corresponding to balanced data is flat.** Percentage improvement in test accuracy over the default batch size of 128 at different training ratios. Experiments conducted on CIFAR-10.

Our findings reveal that while the newer TrivialAugment method exhibits superior performance on balanced training data, the older AutoAugment method yields better results on highly imbalanced data.

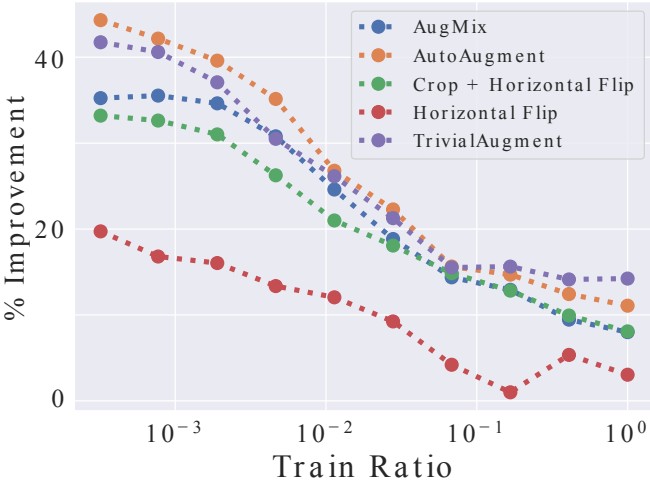

**Figure 6: Optimal augmentations depend on the imbalance ratio.** We plot the percent improvement in test accuracy for different augmentations compared to training without augmentations across train ratios for different augmentations. We see that TrivialAugment, which is known to outperform AutoAugment on class-balanced data, actually performs worse when data is severely imbalanced. Experiments conducted on CIFAR-100.

## A.3 Model architecture

In Figure 8, we illustrate the impact of model size on the performance of the CIFAR-10 dataset with a training ratio of 0.001. The trend observed is similar to the results discussed in the main text, where increasing the model size leads to overfitting in the case of imbalanced training.

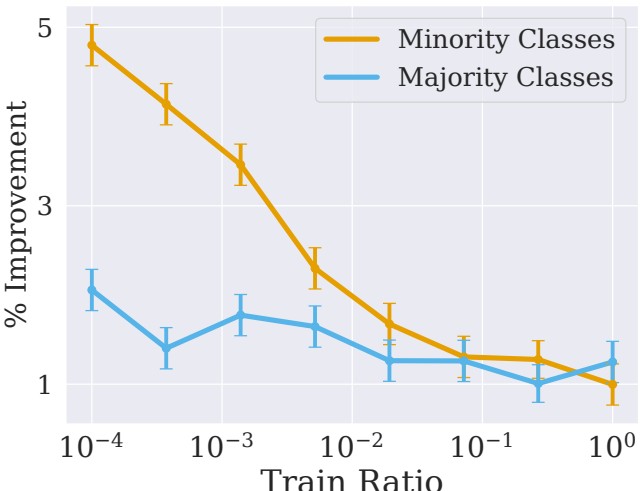

**Figure 7: Strong augmentations are particularly effective at improving minority class accuracy under severe class imbalance.** The percent improvement in test accuracy of TrivialAugment compared to training without any augmentation as a function of the training ratio. Experiments conducted on CIFAR-10. Error bars represent one standard error over 5 trials.

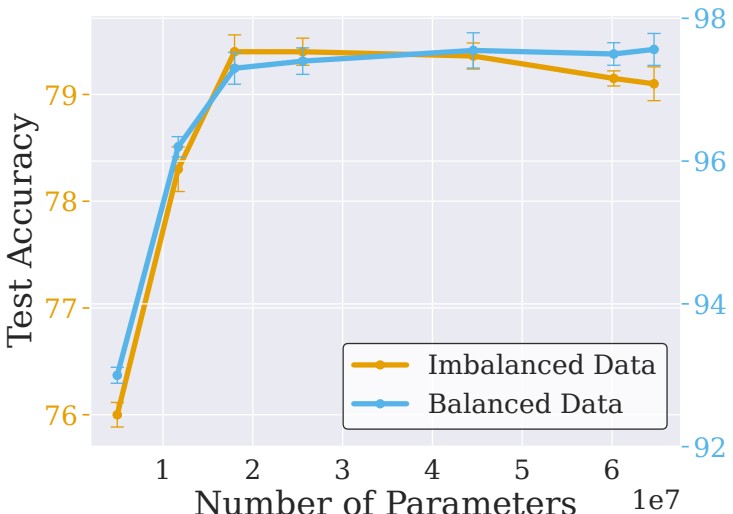

**Figure 8: Bigger architectures overfit on class-imbalanced data.** Experiments conducted on CIFAR-10. Error bars represent one standard error over 5 trials.

### A.4 Pre-training

To assess the effectiveness of pre-training, we fine-tune several pre-trained models on downstream datasets with varying training ratios. In addition to the main body, Figure 10 illustrates the percentage improvement in test accuracy compared to random initialization for supervised pre-training on ImageNet-1k and ImageNet-21k, as well as SimCLR on ImageNet-1k (which is a Self-Supervised Learning (SSL) method), measured by downstream performance on CIFAR-10. This comparison is made across different training ratios (Figure 4). Let $Acc^{\rho}_{\text{Rand}}$ denote the accuracy of the model trained from random initialization at a training ratio $\rho$. The relative improvement is then defined by:

$$\bar{Acc}^{\rho}_b = \frac{Acc^{\rho}_b - Acc^{\rho}_{\text{Rand}}}{Acc^{\rho}_{\text{Rand}}} \tag{2}$$

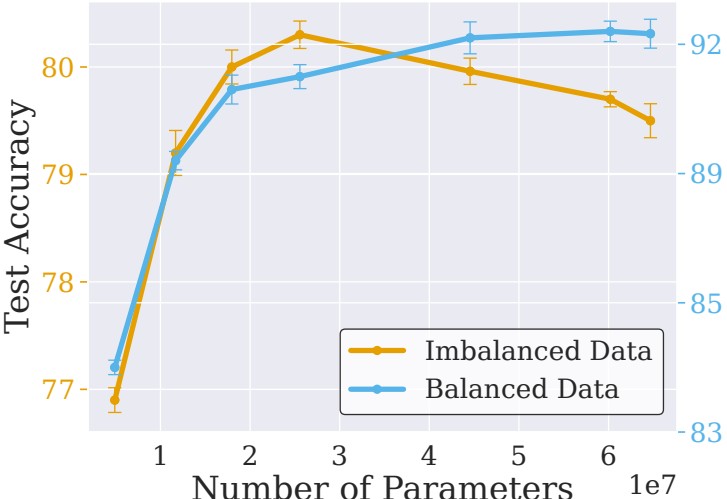

Figure 9: **Bigger architectures overfit on class-imbalanced data.** Experiments were conducted on CINIC-10 with an imbalanced train ratio of 0.001. Error bars represent one standard error over 5 trials.

Positive values indicate an improvement in performance compared to random initialization. It is clear that all pre-training methods improve performance when compared to random initialization. Interestingly, these improvements are significantly more pronounced under imbalanced conditions.

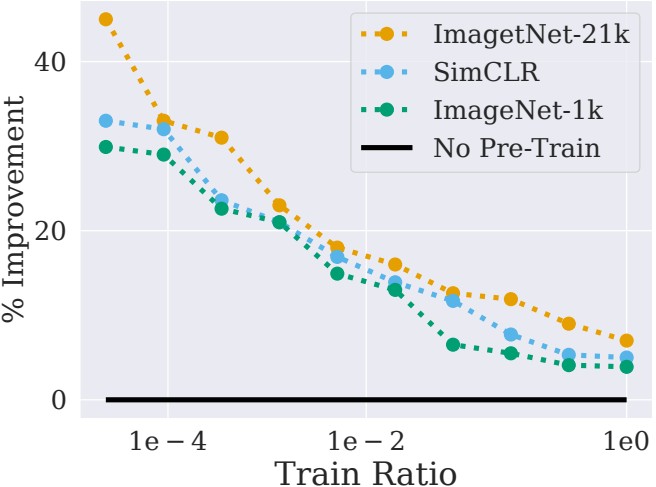

Figure 10: **Pre-training yields bigger improvements on more imbalanced data.** The improvement in the test accuracy compared to training from random initialization. Experiments conducted on CIFAR-10.

## A.5 SSL

Self-supervised learning (SSL) has gained substantial traction as a method of representation learning across multiple domains, including computer vision, natural language processing, and tabular data [12, 40, 65]. Networks pre-trained using SSL often demonstrate more transferable representations than those pre-trained with supervision [24]. Pre-training traditionally consists of a two-stage process: initial learning on an upstream task followed by fine-tuning on a downstream task. However, the limitation in many use-cases is the lack of large-scale pre-training datasets. In order to solve this problem, our approach diverges from this two-stage process by merging supervised learning with an

auxiliary self-supervised loss function during from-scratch training, effectively eliminating the need for any pre-training.

For this, we employ the Variance-Invariance-Covariance Regularization (VICReg) objective [5]:

Given two batches of embeddings, $\boldsymbol{Z} = [f(\boldsymbol{x}_1), \dots, f(\boldsymbol{x}_B)]$ and $\boldsymbol{Z}' = [f(\boldsymbol{x}'_1), \dots, f(\boldsymbol{x}'_B)]$, each of size $(B \times K)$, where $\boldsymbol{x}_i$ and $\boldsymbol{x}'_i$ are two distinct random augmentations of a sample $I_i$, we derive the covariance matrix $\boldsymbol{C} \in \mathbb{R}^{K \times K}$ from $[\boldsymbol{Z}, \boldsymbol{Z}']$.

Consequently, the VICReg loss can be articulated as:

$$\mathcal{L}_{\mathcal{SSL}} = \frac{1}{K} \sum_{k=1}^{K} \left( \alpha \max \left( 0, \gamma - \sqrt{C_{k,k} + \epsilon} \right) + \beta \sum_{k' \neq k} (C_{k,k'})^2 \right) + \gamma \|\boldsymbol{Z} - \boldsymbol{Z}'\|_F^2 / N.$$

In our experiments, the total loss is given by

$$L_{Joint-SSL} = L_{SSL} + \lambda L_{\text{Supervised}}.$$

Note that the SSL loss function is independent of the class-imbalanced labels.

## A.6  SAM

Sharpness-Aware Minimization [19] is an optimization technique that seeks to find "flat" minima of the loss function, often leading to improved generalization. This method consists of taking an initial ascent step followed by a descent step, aiming to find parameters that minimize the increase in loss resulting from the ascent step. Huang et al. [36] demonstrate that flat minima correspond to wide-margin decision boundaries.

Given a model parameterized by weights $\theta$ and a loss function $L(\theta)$ that we aim to minimize, SAM performs two steps in each iteration:

1. **First step (gradient ascent):** Perform a scaled gradient ascent step from the current model weights $\theta$:

$$\theta' = \theta + \rho |\nabla L(\theta)|_2 \frac{\nabla L(\theta)}{|\nabla L(\theta)|_2} \tag{3}$$

2. **Second step (weight update):** Update the weights from $\theta$ in the negative direction of the gradient computed at the post-ascent parameter vector:

$$\theta = \theta - \eta \nabla L (\theta') \tag{4}$$

In the above steps, $\eta$ represents the learning rate, $\rho$ is a hyperparameter determining the size of the neighborhood around the current weights, and $|\cdot|_2$ denotes the Euclidean norm.

SAM was initially developed for balanced datasets, where the decision boundaries for each class have comparable areas. However, this assumption does not hold true for imbalanced datasets. To address this, we adapted SAM for use with class-imbalanced datasets by increasing the flatness specifically for minority class loss terms. We propose a new method - SAM-Asymmetric (SAM-A). Our method adjusts the ascent step size ($\rho$) in SAM's inner loop for minority classes by employing a step size inversely proportional to the classes' proportions.

Let $p_i$ be the proportion of class $i$ in the training set. We define the class-conditional ascent step size as:

$$\rho_i = \frac{\rho}{1 - p_i}, \tag{5}$$

where $\rho$ is a scaling factor.

By doing this, we widen the margins around under-represented classes, potentially improving generalization in imbalanced datasets.

## A.7 Label Smoothing

Label smoothing is a regularization technique often used in training deep learning models. It mitigates the model's excessive confidence in class labels, which can improve generalization and reduce overfitting. However, traditional label smoothing assumes a balanced class distribution, which is not always the case in real-world datasets.

To adapt label smoothing for imbalanced training, we propose a class-conditional label smoothing technique. Instead of using a uniform smoothing parameter $\epsilon$, we use a different $\epsilon_i$ for each class $i$, which is proportional to the inverse of the class's proportion within the dataset.

Let $p_i$ be the proportion of class $i$ in the training set. We define the class-conditional smoothing parameter as:

$$\epsilon_i = \frac{\epsilon}{1 - p_i},\tag{6}$$

where $\epsilon$ is a scaling factor.

We then apply label smoothing as follows. Let $p$ be the model's output probability distribution over $K$ classes, and let $q_i$ be the target distribution for class $i$. The smoothed target distribution is:

$$q_{i,j} = (1 - \epsilon_i)I_{y=j} + \frac{\epsilon_i}{K},\tag{7}$$

where $j \in 1, 2, ..., K$, $y$ is the true class, and $I_.$ is the indicator function.

During training, we minimize the cross-entropy loss between the model's predictions $p$ and the class-conditional smoothed labels $q_i$:

$$L = -\sum_{i=1}^{K} q_{i,y} \log p_y\tag{8}$$

By using class-conditional label smoothing, we apply more smoothing to the minority classes and less to the majority classes, which can help the model generalize better when the class distribution is imbalanced.

## A.8 Data Curation

Common intuition dictates that training on data that is more balanced than the testing distribution can improve representation learning by preventing overfitting to minority samples [25, 10, 27]. In this section, we put that intuition to the test by examining the optimal balance of training data. Moreover, while minority class samples may be scarce, a practitioner may be able to collect additional majority class training samples at will, so we also examine the potentially destructive impact of collecting additional majority samples.

### A.8.1 The Relationship Between Train-Time and Test-Time Imbalance

The literature on training routines for class imbalance in machine learning is filled with methods designed for scenarios in which training data is highly imbalanced but testing data is balanced. However, data encountered during deployment is typically also imbalanced. Therefore, we disentangle training and testing balances and investigate how sensitive models are to discrepancies between the two. This study may be particularly important if one considers collecting training data for a downstream application. Should we gather training data with the same balance we anticipate during testing? How worried should we be if the data we encounter during deployment is more or less balanced than the training data we gathered?

We begin by illustrating three scenarios in Figure 11: (1) identical training and testing ratios, (2) balanced training, and (3) the training ratio with the lowest test error (optimal training ratio). We see that training on data with the same imbalance as the testing data is superior to training on balanced data, and the two strategies only approach equal performance when the testing data becomes balanced. We share additional results over different datasets and models in Figure 20, Figure 21, and Figure 22.

We then plot for each test ratio the corresponding train ratio that results in the lowest test error in Figure 12. If the two ratios are perfectly aligned, then points will lie on the diagonal. Indeed, the points are close to the diagonal, indicating that it is best to train with a very similar imbalance ratio to the test dataset, especially for highly imbalanced testing scenarios.

In these previous experiments, we fixed the size of the training set, but what happens as we gather more and more training data? In Figure 15, we train and evaluate a network on different imbalance ratios across training set sizes, and we plot the misalignment between the train and test ratios, referring to the average distance between the optimal train ratio and the specified test ratio. As the amount of training data increases, we see that the optimal training ratio becomes more and more close to the ratio of the test data.

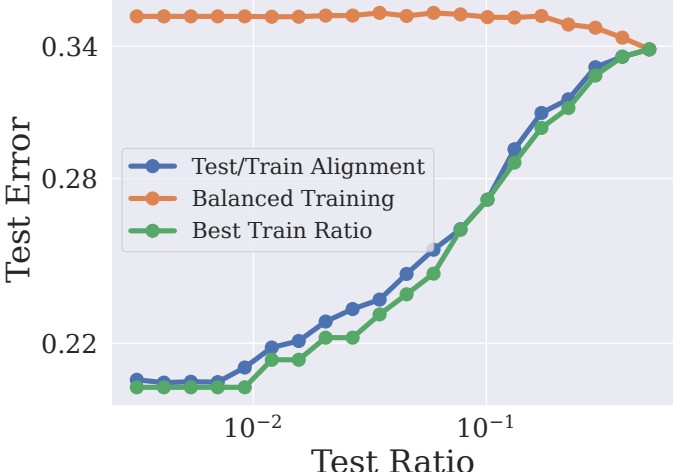

**Figure 11: Imbalanced training data is optimal for imbalanced testing scenarios.** Test accuracy as a function of the test ratio for different training setups. Experiments conducted on CIFAR-100.

### A.8.2 When More Data Degrades Performance

In practice, a practitioner may not have precise control over the data they collect. Will collecting additional samples always help performance? Instead of fixing the total number of samples and

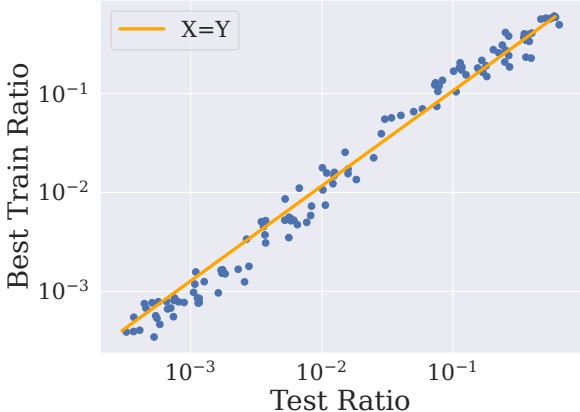

**Figure 12: The optimal train ratio is closely aligned with the test ratio.** Experiments conducted on CIFAR-100.

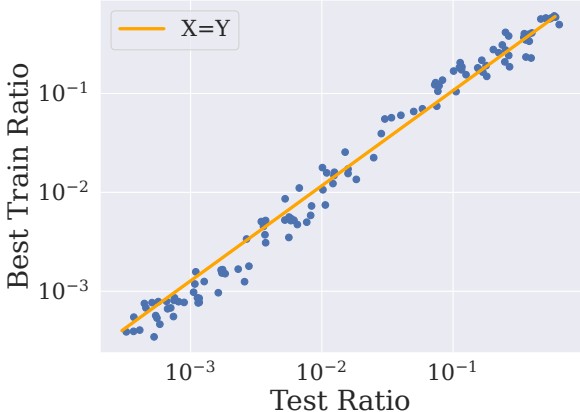

**Figure 13: The optimal train ratio is closely aligned with the test ratio.** Experiments conducted on CINIC-10.

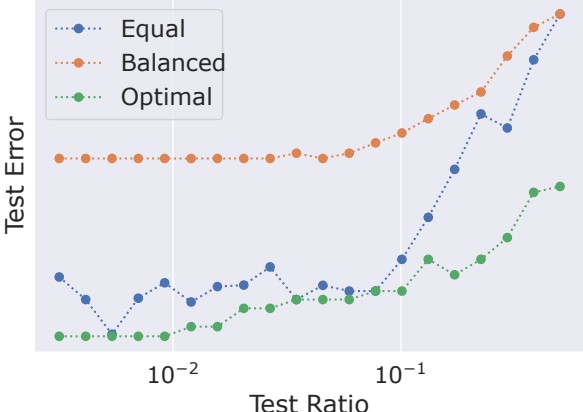

**Figure 14:** Test accuracy on the minority classes as a function of the test ratio for different training setups. 'Equal' denotes the same balance between training and testing, and 'Optimal' is the optimal trainset balance amongst the ratios we try. Experiments conducted on CIFAR-10.

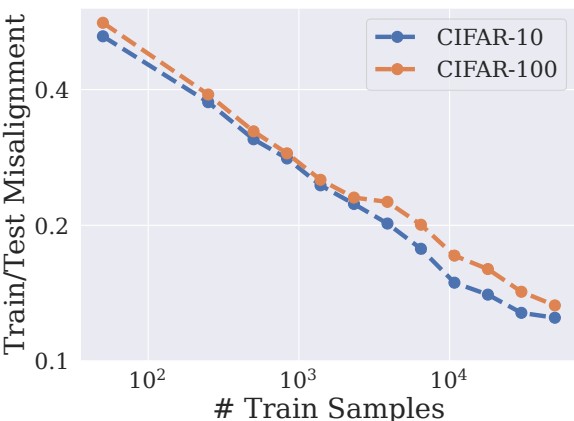

**Figure 15: Alignment between train and test proportions improves as the number of training samples increases.** *Train/test misalignment* is calculated by taking the mean over test ratios of the difference between the best train ratio (train ratio that gives maximum test accuracy) and the test ratio. If misalignment is 0, then the best train ratio is always the same as the test ratio.

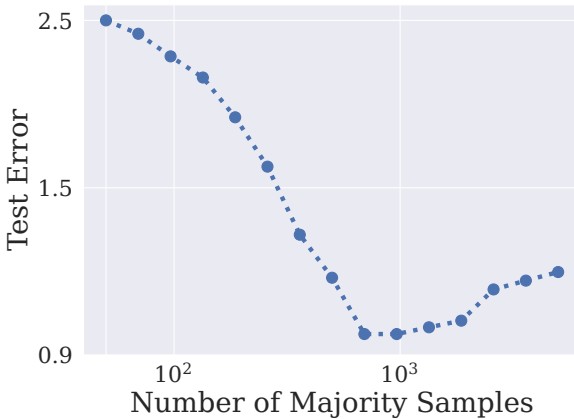

**Figure 16: The potentially destructive effects of adding majority class samples**. We fix the number of minority samples to be 200 and vary the number of majority samples. Experiments conducted on CIFAR-100.

**Table 5: Our training routines exceed previous state-of-the-art or improve existing methods when combined.** Split class accuracy for classes with Few, Med and Many examples of WideResNet-28×10 on long-tailed CIFAR-100 and CINIC-10. Error bars correspond to one standard error over 5 trials.

| | CINIC-10 | | | CIFAR-100 | | |
|---|---|---|---|---|---|---|
| Method | Few | Med | Many | Few | Med | Many |
| ERM | $40.5 \pm 0.4$ | $64.1 \pm 0.3$ | $90.1 \pm 0.5$ | $20.1 \pm 0.3$ | $42.3 \pm 0.3$ | $70.5 \pm 0.6$ |
| Reweighting | $36.6 \pm 0.5$ | $63.1 \pm 0.3$ | $87.8 \pm 0.3$ | $17.1 \pm 0.4$ | $39.3 \pm 0.3$ | $67.1 \pm 0.4$ |
| Resampling | $37.4 \pm 0.5$ | $63.6 \pm 0.6$ | $87.9 \pm 0.4$ | $18.4 \pm 0.2$ | $38.1 \pm 0.2$ | $68.9 \pm 0.3$ |
| Focal Loss | $39.1 \pm 0.2$ | $63.9 \pm 0.2$ | $88.2 \pm 0.5$ | $19.8 \pm 0.4$ | $39.0 \pm 0.5$ | $69.3 \pm 0.6$ |
| LDAM-DRW | $40.1 \pm 0.4$ | $64.3 \pm 0.4$ | $89.8 \pm 0.3$ | $20.8 \pm 0.5$ | $42.1 \pm 0.3$ | $70.6 \pm 0.4$ |
| M2m | $42.8 \pm 0.7$ | $64.1 \pm 0.6$ | $90.3 \pm 0.4$ | $20.1 \pm 0.6$ | $41.8 \pm 0.4$ | $69.4 \pm 0.5$ |
| SAM-A | $43.2 \pm 0.3$ | $62.3 \pm 0.6$ | $89.7 \pm 0.3$ | $22.5 \pm 0.4$ | $40.3 \pm 0.3$ | $70.1 \pm 0.4$ |
| Joint-SSL + SAM-A | $43.9 \pm 0.4$ | $63.3 \pm 0.5$ | $90.4 \pm 0.5$ | $22.9 \pm 0.3$ | $41.3 \pm 0.6$ | $69.9 \pm 0.6$ |
| Joint-SSL + SAM-A + M2m | $44.1 \pm 0.3$ | $64.2 \pm 0.4$ | $90.9 \pm 0.3$ | $23.9 \pm 0.4$ | $42.3 \pm 0.2$ | $70.4 \pm 0.3$ |

varying their imbalance ratio, we now fix the number of samples from the minority class and vary the number of total majority class samples.

In Figure 16, we see that increasing the number of samples from the majority class initially boosts performance on a balanced test set. Nevertheless, in both cases, the performance reaches an optimum before the growing training data imbalance eventually degrades test accuracy. Thus, adding training data can help, but if we add enough majority samples, we must be careful not to cause too sharp a mismatch between training and testing distributions. Notably, the optimal training set ratio is nearly balanced, matching the test set, even when we are allowed to gather extra samples from one class without having to forego samples from another.

## A.9 Benchmarking Results

In Table 5, we present additional experimental results that compare the methods proposed in our paper with the baseline methods. In accordance with Kang et al. [38], we also report the accuracy across three distinct subsets: (1) Many-shot classes, which contain more than 100 training samples. (2) Medium-shot classes, comprising 20 to 100 samples, and (3) Few-shot classes, including classes with fewer than 20 samples.

**Table 6: SAM-A, our modified label smoothing, and small batch sizes improve performance on class-imbalanced tabular datasets.**

| Method | Otto | Adult | CoverType |
|---|---|---|---|
| XGBoost | 82.7 | 87.5 | 96.9 |
| MLP | 83.0 | 87.4 | 97.5 |
| ResNet | 82.5 | 87.4 | 97.5 |
| FT-Transformer | 82.3 | 87.3 | 97.5 |
| MLP w/ SAM-A + Smoothing | **83.2** | **87.6** | **97.6** |

### A.9.1 Tabular Datasets

**Training procedure** - Our tuning, training, and evaluation protocols are consistent with those in Gorishniy et al. [23]

Accordingly, we tune each model's hyperparameters for each dataset on a validation split. We use the Optuna library [2] to execute Bayesian optimization via the Tree-Structured Parzen Estimator algorithm. In our evaluation process, we run 15 experiments for each tuned configuration using different random seeds, and report the average performance on the test set.

## A.10 Regularization and Overfitting

In order to determine whether the performance differences among various methods stem from their optimization abilities or their generalization to unseen test samples, we evaluate the training error without any regularization or specialized optimization method. Specifically, we train a ResNet-50 network on CIFAR-10 and CIFAR-100 datasets using SGD with an initial learning rate of 0.5 and cosine annealing, across different levels of training data imbalance. As seen in Figure 17, although fitting all training examples takes longer as we increase the imbalance ratio of our datasets, the empirical risk minimization successfully fits all training data eventually, including minority samples.

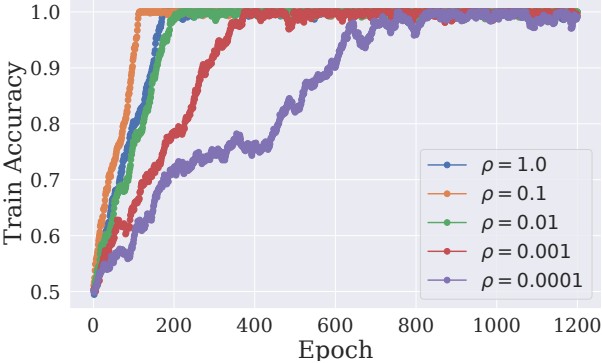

**Figure 17: Imbalanced data is harder to fit.** Training accuracy every epoch for imbalanced training with various imbalance ratios. Experiments conducted on CIFAR-10.

## A.11 Decision Boundary Visualizations

To explore the differences between classifiers trained on imbalanced data, we visualize their decision boundaries. A variety of methods have been established for visualizing the decision boundaries of deep learning models, offering valuable insights into their intricate internal operations. Apart from the methods discussed in the main text, we utilize the approach introduced by Somepalli et al. [66] to visualize the decision boundaries of a ResNet-50 network trained on the CIFAR-10 dataset. In Figure 19, we display the decision boundaries resulting from standard training (right), which yields

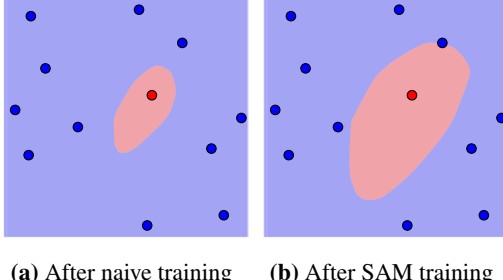

**(a)** After naive training  **(b)** After SAM training

**Figure 18: SAM-A pulls decision boundaries away from minority samples, whereas standard training routines overfit.** Experiments conducted on a toy 2D classification problem with a multi-layer perceptron.

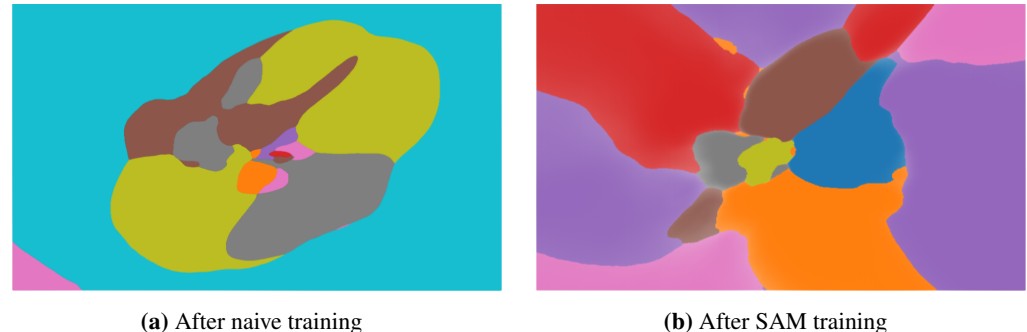

**(a)** After naive training  **(b)** After SAM training

**Figure 19: SAM-A makes decision regions take up similar volumes, whereas standard training routines shrink wrap the decision boundaries around minority samples.** Experiments conducted on a CIFAR-10 with ResNet-18.

narrow margins around minority classes (green, grey, and orange), and SAM-A (left), which notably broadens these margins and all the classes occupy similar area in input space.

## B Experimental Details

In this section, we provide additional implementation details that were not included in the main text.

For the CIFAR-10, CIFAR-100, and CINIC-10 datasets, we follow the imbalanced setup proposed by Liu et al. [54], using an exponential distribution to create imbalances between classes. Across all methods, we use TrivialAugment [58] combined with CutMix as our augmentation policy, supplemented by label smoothing and an exponential moving weight average. Our model of choice is the WideResNet-28×10 [73].

We employ the SGD optimizer with momentum 0.9 and weight decay coefficient $2 \times 10^{-4}$. Our models are trained for 300 epochs with cosine annealing and a linear warm-up of the learning rate. The learning rate is initialized at 0.1.

For the APTOS 2019 Blindness Detection, SIIM-ISIC Melanoma Classification, and EuroSAT datasets, we largely follow the approach detailed in Fang et al. [18], utilizing the ResNeXt-50-32×4d model, which was identified as the best model for these datasets in the comparison by Fang et al. [18].

Our implementation was done in PyTorch, utilizing the PyTorch Lightning library for training. All of our models were trained on V100 GPUs.

## C Limitations

In our paper, we found that existing methods for class imbalance are unreliable on real-world datasets. While our tuned routine was effective on the real-world datasets we considered, these general trends

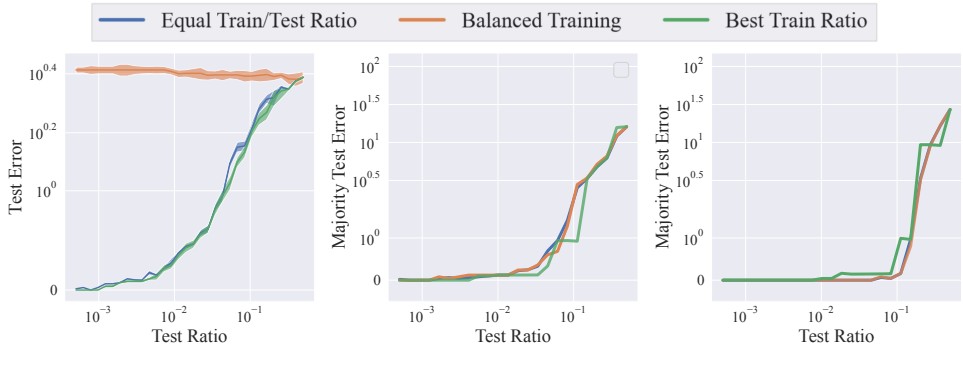

**(a)** ResNet on CIFAR-10 Dataset **(b)** XGBoost on Adult Dataset **(c)** SVM on Forest Cover Dataset

**(d)** ResNet on CIFAR-10 Dataset **(e)** XGBoost on Adult Dataset **(f)** SVM on Forest Cover Dataset

**Figure 20:** Test error split by majority and minority classes for balanced test sets. We see similar trends across all models and datasets.

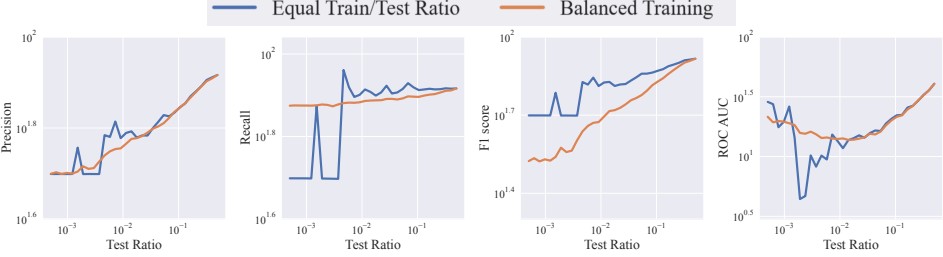

**Figure 21:** Additional metrics for XGBoost on the Adult dataset.

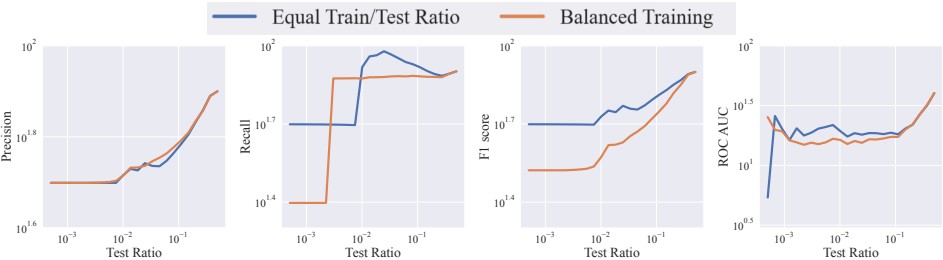

**Figure 22:** Additional metrics for SVM on the Forest Cover dataset.

raise the concern that solutions which are effective on some class-imbalanced datasets may fail on others. A second limitation of our work is that some tools we utilize are only applicable in certain domains. For example, data augmentations and self-supervised learning for tabular data are not widely accepted.

## D   Broader Impacts

Across a wide variety of high-impact domains, ranging from credit card fraud detection to disease diagnosis, data is severely class-imbalanced. Therefore, performance increases for class-imbalanced data is highly valuable. With this potential for value also comes the potential that proposed methods make false promises which won't benefit real-world practitioners and may in fact cause harm when deployed in sensitive applications. For this reason, we release our numerical results across diverse datasets, and we also include implementation details for the sake of transparency and reproducibility. As with all new state-of-the-art methods, our improvements may also improve models used for malicious intentions.

