# OpenReview forum: "Simplifying Neural Network Training Under Class Imbalance"
_NeurIPS.cc/2023/Conference — NeurIPS 2023 poster_

### Official Review · Reviewer_E6os · 2023-06-08

**Soundness:** 4 excellent
**Presentation:** 4 excellent
**Contribution:** 3 good
**Rating:** 8
**Confidence:** 5

**Summary:**

The authors show that simply tuning standard hyperparameters provides state-of-the-art performance on a wide variety of class-imbalanced datasets, which may be surprising and give an impact to the community: We have to re-think the experimental settings for performance evaluation on imbalanced datasets.

The authors show that simply tuning existing components of DNNs, such as the batch size, data augmentation, architecture size, pre-training, optimizer, and label smoothing, can achieve state-of-the-art performance without any specialized loss functions or samplers.
Specifically, (1) imbalanced data prefers small batch sizes, (2) the data augmentation strategies that achieve the best performance on balanced datasets yield inferior performance on imbalanced datasets, (3) large models that do not overfit to balanced datasets strongly overfit to imbalanced datasets of the same size, (4) adding an self-supervised loss during training can improve the performance on imbalanced datasets, (5) the SAM optimizer improves minority accuracy, and (6) label smoothing prevents the overfitting to minority classes.

Thorough experiments are performed on six image datasets, including natural image, medical, and remote sensing datasets as well as two tabular datasets. The models are a variety of CNNs, XGboost, and SVM. The authors run five seeds for each evaluation and report the mean and one standard error. The performance improvement compared with state-of-the-art (e.g., [Zhou+, CVPR2023, http://home.ustc.edu.cn/~zzp1994/2023068462.pdf]) is significant (especially on CIFAR-10 and -100).

The authors conclude that, even though they attain the state-of-the-art, existing methods designed for web-scraped natural image classification benchmarks do not always yield improvements on other real-world problems. If we are to reliably compare methods for class imbalance, we need a more diverse benchmark suite, i.e., there may not a universally good model for imbalanced datasets.

The paper is well-written and easy to follow. I enjoyed reading the paper. Contribution is clear. Experimental settings are well-explained. Distinctions from previous works are clearly stated. Overall, the contribution and impact of the paper is significant compared with other papers that have been published in top conferences.

**Strengths:**

- The authors show that simply tuning standard hyperparameters provides state-of-the-art performance on a wide variety of class-imbalanced datasets, which may be surprising and give an impact to the community: We have to re-think the experimental settings for performance evaluation on imbalanced datasets.

- Experimental results are convincing. Thorough experiments are performed on six image datasets, including natural image, medical, and remote sensing datasets as well as two tabular datasets, which are sufficient compared with other contemporary papers published top conferences. The models are a variety of CNNs, XGboost, and SVM. The authors run five seeds for each evaluation and report the mean and one standard error. The performance improvement compared with state-of-the-art (e.g., [Zhou+, CVPR2023, http://home.ustc.edu.cn/~zzp1994/2023068462.pdf]) is significant (especially on CIFAR-10 and -100).

- The paper is well-written and easy to follow. Contribution is clear. Experimental settings are well-explained. Distinctions from previous works are clearly stated. Overall, the contribution and impact of the paper is significant compared with other papers that have been published in top conferences.

- There are some papers that claims that supervised contrastive learning is effective for imbalanced learning, e.g., [Wang+, CVPR2021, https://arxiv.org/abs/2103.14267]. There is also a seminal paper that analyze self-supervised learning and class-imbalance [44], in which Liu+ claim that pre-trained self-supervised representations are more robust to imbalance than supervised representations. To my understanding, the authors' approach is somewhat similar to (but not exactly the same as) these two, i.e., adding self-supervised loss to supervised training and mitigating the pre-training phase (Joint-SSL). The authors show that this approach is effective on imbalanced datasets, which is a consistent result with the two references above, although Joint-SSL only is not always helpful or the improvement is marginal (Tables 1 & 2).

**Weaknesses:**

- In this paper, hyperparameter tuning is shown to be important for performance on imbalanced datasets. THEREFORE, I would like to see ALL the hyperparameter settings and tuning method of ALL the experiment. I would like to see the code to reproduce the experimental results.

**Questions:**

- [Question] Could you discuss the differences of underlying mechanism of Joint-SSL and supervised contrastive learning, e.g., [Wang+, CVPR2021, https://arxiv.org/abs/2103.14267]? (I know the difference of the loss functions between the two.) This is a bit abstract question, but I would appreciate it if you could answer this.

- [Question] It is shown that SAM and its variant are effective on imbalanced datasets in [Azzuni+, ISBIC2022, https://ieeexplore.ieee.org/document/9854725] and [Zhou+, CVPR'23, http://home.ustc.edu.cn/~zzp1994/2023068462.pdf].
Could you briefly discuss the relationships between the present paper and these two?

- [Question] In [Shen+, ICLR2021, https://openreview.net/pdf?id=PObuuGVrGaZ], they discuss in Section 7 that label smoothing is ineffective when the dataset is imbalanced in the context of knowledge distillation. Could you discuss the relationships between this result and 6. in Introduction of the current paper?

- [Comment] On the other hand, label distribution smoothing in imbalanced regression problems is shown to be effective in [Yang+, ICML2021, http://dir.csail.mit.edu/]. There may be something more in label smoothing and imbalance. Interesting.

- [Question] Could you explain why are smaller batch sizes beneficial for more imbalanced datasets? The authors discuss this point: there is a tradeoff between (1) large batch sizes prevent from forgetting minor classes by including minority samples in almost every batch and (2) large batch sizes lead to overfitting to majority classes because they dominate almost all batches. If so, what is the main factor that controls this tradeoff? This would be a difficult question, and just a guess is welcome.

- [Comment (minor)] (Related to Figure 1) I would like to see more results on different datasets and models other than CIFAR-10 and -100 with ResNet, if possible.

- [Question (minor)] I know the focus of the present paper is CNNs, but are there any experimental results of Transformer-based models? I wonder if there is any differences between CNNs and Transformers.

- [Comment (major)] In this paper, hyperparameter tuning is shown to be important for performance on imbalanced datasets. THEREFORE, I would like to see ALL the hyperparameter settings and tuning method of ALL the experiment. I would like to see the code to reproduce the experimental results. This is also because it seems that several specialized methods are needed to achieve SOTA such as TrivialAugment + CutMix + label smoothing + exponential moving weight average + Joint-SSL + SAM-A (+ M2m) (please correct me if it is wrong). In addition, batch size, choice of data augmentation methods, learning rate, and other hyperparameters are also tuned, I guess.

- [Comment (minor)] Optimizer dependence of the classification performance on imbalanced datasets is also interesting.

**Limitations:**

- As stated by the authors, the optimal learning configuration depends on the task, and thus it may be unclear how universal the know-how obtained in this paper is.

---

> ### Author Rebuttal · Authors · 2023-08-09
>
> Dear Reviewer nEkH,
>
> Thank you for your thorough and insightful feedback. We are grateful for your recognition of our work's significance and potential impact. We address each of your points below:
>
> ## Hyperparameter Settings and Code Availability:
> Regarding hyperparameter tuning, we adopt a standard process as reported in previous works [1]. In our revised working draft, we now provide a more comprehensive appendix detailing all hyperparameter settings. Furthermore, we will make our code publicly available upon publication to ensure full reproducibility of our results.
>
> ## Compare to Supervised Contrastive Learning:
> In our work, Joint-SSL combines self-supervised and supervised losses, aiming to leverage the benefits of both approaches. Unlike supervised contrastive learning, even if every sample was in the same class, the SSL objective allows us to learn useful features.  Another difference lies in the choice of self-supervised loss: In our implementation, we use a non-contrastive objective - VICReg - along with the supervised loss, while supervised contrastive learning uses a contrastive loss. We appreciate your comment and clarify this in our revised version.
>
> ## Relationships with SAM:
> Thank you for pointing out these papers. We can't find the Azzuni+ paper and would appreciate a pointer. [1] suggest using SAM directly to combat data imbalance without specifying optimizing the perturbation sizes per class. However, [2], whose work was parallel to ours, suggests a more complicated version of SAM requiring a two-stage process. We agree that the first step of their method is similar to ours, and we will update our revised paper accordingly.
>
> ## Label Smoothing and Imbalance:
> Shen et al. (2021) found that smoothing the same for all classes is ineffective for imbalanced datasets. However, our suggested method applies different smoothing for each class based on their distributions. We apply more smoothing to the minority classes, which need regularization. This is consistent with our other findings that methods increasing regularization and robustness against overfitting of the minority classes are beneficial for imbalanced training.
>
> ## Why Smaller Batch Sizes are Beneficial for More Imbalanced Datasets:
>
> Our current understanding is that modern DNNs tend to perform well with large batch sizes during balanced training. This is partly because larger batches inherently provide a form of implicit regularization. They include diverse samples from different classes in each batch, creating heterogeneity in the data that promotes generalization. On the other hand, this heterogeneity is often lacking for imbalanced training. Therefore, with large batches, DNNs may overfit to minority classes. Conversely, smaller batch sizes introduce a form of regularization. Specifically, we found that smaller batch sizes help prevent overfitting minority classes. This acts as a form of regularization in SGD as described by Sekhari et al. [3], who analyzed the role of batch size as a regularizer for the loss landscape, encouraging generalization. This effect introduces noise into gradient estimation and counteracting the tendency to overfit to minority examples.
>
> To delve deeper into this issue, we examine its interaction with our joint-SSL method. While SSL models typically perform better with larger batch sizes for balanced datasets, our study suggests that the conventional wisdom for SSL models may not apply to imbalanced data. To verify this, we trained our joint-SSL algorithm with different batch sizes on CIFAR-10. As seen in the provided page, the impact of batch size is less pronounced, with joint-SSL acting as an alternative regularizer, mitigating the overfitting of minority classes.
>
>
> We suspect the heterogeneity of samples might control the trade-off between including more samples in each batch and avoiding overfitting. However, we acknowledge that this is a complex topic, requiring further investigation to understand the underlying mechanism fully.
>
> ## Additional Results on Different Datasets and Models:
> Prompted by your feedback, we have included additional results with the Tiny-Imagnet and iNaturalist, using Swin Transformer v2 and ConvNeXt. The results of these additional experiments consistently align with our previous findings, enhancing our work's robustness and general applicability. Furthermore, these enhancements increase the diversity of our architectural analysis, contributing to a broader understanding of various architectures' behavior under imbalance.
>
>
> ## Experimental Results of Transformer-Based Models:
> While our focus in this paper is primarily on CNNs, in response to your comment, we have analyzed the transformer-based model, Swin Transformer v2. As you can see in the attached figure, the resulting behavior is similar to that exhibited in CNN experiments.
>
> ## Optimizer Dependence on Classification Performance:
> We have now added a more rigorous analysis to our revised working draft, and we’ll include it in our camera ready version. In brief, the differences between SGD and AdamW are small. Generally, while AdamW was better for larger batch sizes and more robust to hyperparameter choices, SGD achieved better results for imbalanced training.
>
> Thank you again for your thoughtful review. We made a significant effort to address your feedback, including multiple experiments and paper edits, and we would appreciate it if you would consider raising your score in light of our response.  Please let us know if you have additional questions we can address.
>
> ## References:
> [1]  Liu et al., 2021, “Self-supervised learning is more robust to dataset imbalance”.
>
> [2] Zhou et al., 2023, “Class-Conditional Sharpness-Aware Minimization for Deep Long-Tailed Recognition”.
>
> [3]  Sekhari et al., 2021, "Sgd: The role of implicit regularization, batch-size and multiple-epochs”.

---

> > ### Comment · Reviewer_E6os · 2023-08-14
> >
> > Thank you for the discussions and thorough revisions.
> > I really appreciate the significant effort.
> > I carefully read all the comments (and other reviews).
> > Let me give some more comments below.
> >
> >
> > > We can't find the Azzuni+ paper and would appreciate a pointer.
> >
> > I am sorry for the inconvenience.
> > Azzuni+ is ["Color Space-based HoVer-Net for Nuclei Instance Segmentation and Classification", Azzuni+, IEEE ISBICC 2022, https://arxiv.org/abs/2203.01940].
> >
> >
> > ### Why Smaller Batch Sizes are Beneficial for More Imbalanced Datasets:
> > Thank you for the detailed reply.
> > This is also pointed out by Reviewers kUXT & 8NJC, but your explanation is convincing.
> > My understanding is as follows.
> > A large batch size leads to small variance of gradients.
> > The batch is likely to be dominated by major examples.
> > Thus, the weight tends to converge to a local minima that prefers majority class.
> > On the other hand, a small batch size induces noise to gradients, which perturbs the gradients and thus they do not always go to the poor local minima.
> >
> >
> > ### Related to Question on Batch Size and SSL [Reviewer kUXT]:
> > Interestingly, a recent study states that large batch sizses are not always necessary for good performance [1].
> > Is this relevant to the present paper?
> >
> > [1] ON THE DUALITY BETWEEN CONTRASTIVE AND NON- CONTRASTIVE SELF-SUPERVISED LEARNING
> >
> >
> > ### I also would like to see results on transformers [also by Reviewer kUXT & 8NJC]:
> > I think the lack of it is not a critical problem
> > because major CNN models are included, and thus, the paper provide many insights to CNN users.
> > Anyways, in Author Response, the authors added extra results on SwinV2 (and ConvNext, additional datasets, etc.),
> > which addressed the concern.
> >
> >
> > ### Language models, imbalance, and batch size:
> > Related to future work, [2] analyzes multilingual models and shows that minority language are affected by imbalance. Oversampling is not that effective in this scenario.
> > The proposed method is shown to be effective even when the batch size is large.
> >
> > [2] Robust Optimization for Multilingual Translation with Imbalanced Data
> >
> >
> > ### Lesson 6:
> > I found two relevant papers related to Lesson 6.
> > [3] uses different hyperparameters for different classes to address the class-imbalance problem.
> > [4] uses class-dependent temperature for imbalanced learning.
> > Relations, differences, and discussions should be included in comparison with Lesson 6.
> >
> > [3] AutoBalance: Optimized Loss Functions for Imbalanced Data
> >
> > [4] Identifying and compensating for feature deviation in imbalanced deep learning ([61] in the main text).
> >
> >
> > ### Contributions Clarified:
> > The contributions are clarified more in Author Response to Reviewer 8NJC.
> >
> >
> > ### Some of the findings are valid for balanced setup as well [Reviewer kUXT,]:
> > This is a comment by Reviewer kUXT,
> > but it is important to show that the techniques that are effective in balanced setup are effective also in imbalanced setup.
> > This is because not all techniques work well universally.
> > In particular, whether a sampling or augmentation method is effective in imbalanced training is not obvious in my opinion;
> > for instance, when to use oversampling and undersampling is still an open question.
> > Therefore, this point is not critifcal and I still support the acceptance.
> >
> >
> > # [Important] Evaluation Metrics [Reviewer D6vk]:
> > I thank D6vk for pointing this out.
> > Some of the results are given in classwise accuracy, but some are given in accuracy,
> > which is notoriously known to prefer major classes (and confusingly, some papers use term "accuracy" for "(averaged) classwise accuracy"!).
> > Thus, I personally do not recommend you to use accuracy in imbalanced learning.
> > Instead, I recommend the macro-averaged recall (averaged classwise accuracy), macro-averaged AUC, and macro- and micro-averaged F1 score, for example.
> > In view of this point, are there any potential changes in the conclusion, or can you include more metrics?
> >
> > [5] is a comprehensive, nice study on binary confusion matrix-based metrics and provide more detailed differences between accuracy, recall, F1, precision, specificity, and so on.
> >
> > [5] The Impact Of Class Imbalance In Classification Performance Metrics Based On The Binary Confusion Matirx

---

> > > ### Author Response · Authors · 2023-08-19
> > > **Second Response for Reviewer E6os - Part 1**
> > >
> > > Dear Reviewer E6os,
> > >
> > > Thank you once again for your thoughtful and comprehensive feedback, and for the additional references you provided. We appreciate the time you have invested in our work. Below, we address each of your points in detail:
> > >
> > > **Azzuni+ Paper Reference:**
> > > Thank you for providing the reference to the Azzuni+ paper. Unlike Azzuni et al., who suggest using SAM directly to combat imbalance without optimizing the perturbation per class, our modified SAM treats majority and minority classes differently, adjusting the decision boundaries specifically for minority classes.
> > >
> > > **Batch Sizes and Imbalanced Datasets:**
> > > Your explanations are accurate. To further investigate this issue, we conducted additional experiments, which are detailed in the general comment section. In summary, we found that smaller batch sizes can help mitigate overfitting, especially for minority classes. First, we analyzed the train/test errors for both minority and majority classes on Tiny-ImageNet using ConvNetXt, finding that minority classes tend to overfit significantly more with larger batch sizes; this overfitting decreases as batch size reduces, suggesting smaller batch sizes can help mitigate overfitting for minority classes. Second, we examined the variance of neural network parameter gradients, comparing balanced and imbalanced training scenarios. Our results show that imbalanced training results in lower gradient noise, particularly with larger batch sizes, which correlates with a higher propensity for models to overfit under imbalanced training conditions. Lastly, we investigated loss flatness through the Hessian spectrum, a well-known indicator of generalization and overfitting. Our findings reveal that, in imbalanced training, larger batch sizes lead to convergence at points with higher loss curvature, as characterized by the leading eigenvalue of the Hessian.
> > >
> > > **Contrastive and Non-Contrastive SSL:**
> > > Thank you for pointing out the paper. This paper claims that with more tuning of the InfoNCE parameters, the robustness of SimCLR and MoCo to small batches can be improved. Our hypothesis is that, as we mentioned before, for small batch sizes, along with serving as a regularizer, there is more noise in the gradient estimation which may lead to bad minima (especially in SSL, where the signal is weak compared to supervised learning). However, there are more ways to prevent high noise than increasing the batch size. We believe that exploring methods that combine the advantages of small batch sizes, which prevent overfitting, with solutions to the noise problem for these small batch sizes, is a very interesting future research direction.
> > >
> > > **Transformer Results:**
> > > We are pleased to hear that you found our additional experiments, including those on the Swin Transformer V2, to be satisfactory. We believe these results strengthen our paper's insights into various architectures under class imbalance.
> > >
> > > **Language Models, Imbalance, and Batch Size:**
> > > We appreciate the pointer to the work analyzing multilingual models under imbalance. This is indeed an interesting and relevant direction. We would like to point out that in language models (and actually in every domain where natural labels are not available), there may be several kinds of imbalanced data, not only across different languages. For example, there can be imbalances in different styles or structures of language. This phenomenon makes research on imbalanced data in language models much more complicated, but also very interesting, of course.
> > >
> > > **Lesson 6 and Related Papers:**
> > > Thank you for highlighting these papers. [3] indeed uses different hyperparameters for different classes to address the class imbalance. However, this method is much more complicated and requires extensive hyperparameter search based on the validation loss, and it even includes special data augmentation techniques. Moreover, as mentioned in our paper, there are many works which modify the loss function to behave differently for majority and minority classes. The main advantage of our label smoothing method is its simplicity and that it doesn't require specific hyperparameters designed specially for imbalanced data. Regarding [4], thank you for pointing out this paper. We agree that the class-dependent temperature technique is highly relevant to our work and can be seen as a different way of "smoothing" the output of the network. We believe that the main advantage of our smoothing method is its similarity to the current label smoothing method, which is common in balanced learning. In this case, you don't need to investigate this method each time for imbalanced training; instead, you can apply knowledge gained from balanced training.  We now discuss these works in our updated working draft and appreciate your suggestions.

---

> > > > ### Author Response · Authors · 2023-08-19
> > > > **Second Response for Reviewer E6os - Part 2**
> > > >
> > > > -->
> > > >
> > > > **Effectiveness of Techniques in Balanced and Imbalanced Setups:**
> > > > We completely agree with your point. Our work emphasizes that not all techniques that work well in balanced setups are equally effective in imbalanced setups. This is a critical aspect, and we will ensure that our revised manuscript makes this clear. Moreover, we believe the value of our paper lies in the new perspectives we provide on class-imbalanced training. We demonstrate the benefits of our simple and easy-to-use tuned or imbalance-adapted training techniques. These are specifically designed for the context of class imbalance, a field that was previously dominated by specialized loss functions or samplers designed specifically for class imbalance. Our work represents a perspective shift on training under class imbalance, and we have achieved significant performance gains across multiple benchmark and real-world datasets.
> > > >
> > > > **Evaluation Metrics:**
> > > > As we mentioned in our response to Reviewer D6vk, in our paper we reported both accuracy, due to its simplicity and direct interpretability, and class-wise accuracy in the main text and in the appendix. We will make sure to provide the averaged class-wise accuracy in the cases that don't currently appear in the paper.
> > > > Prompted by this feedback, we have also incorporated the test likelihood error, which helps to assess the model's probabilistic performance (see the attached page). Additionally, we will report the macro-averaged AUC and both macro- and micro-averaged F1 scores for all our experiments. These new metrics exhibit behavior similar to the aforementioned metrics and we will incorporate them in our camera-ready version of our paper.
> > > >
> > > > Thank you once again for your constructive and detailed feedback. Your insights have been instrumental in identifying areas where our paper can be improved. We have made every effort to address your points comprehensively in our revised manuscript through extensive experiments and clarifications. We kindly request that you consider these revisions in your final assessment and consider raising your score in light of our response. Please do not hesitate to let us know if there are additional points we can address.

---

> > > > > ### Comment · Reviewer_E6os · 2023-08-21
> > > > >
> > > > > Thank you for the quick, detailed, and convincing response. I carefully read then and found they make sense.
> > > > > The additional experiments are relevant and valuable, and I believe they help the readers to understand the mechanism of imbalanced training much better.
> > > > > All of the questions and concerns, in particular, about the evaluation metrics, are now fully addressed.
> > > > > I strongly support the acceptance of the paper.
> > > > >
> > > > > My rating and confidence score are kind of saturated, so I raised Soundness.
> > > > > Again, I enjoyed reading the paper, and thank you for your great effort!

---

### Official Review · Reviewer_nEkH · 2023-06-29

**Soundness:** 3 good
**Presentation:** 3 good
**Contribution:** 2 fair
**Rating:** 6
**Confidence:** 4

**Summary:**

The paper studies the long-tail recognition problem and the impact of existing components of standard deep learning pipelines on the generalization performance, such as the batch size, data augmentation, architecture size, pre-training, optimizer, and label smoothing. They find that simply tuning those components can achieve state-of-the-art performance without any specialized loss functions or samplers.

**Strengths:**

- Rethinking the effect of existing components in long-tailed recognition is a very important research topic.
- It is interesting to show that the minority class performance decreases at some point with the increase of the model size for imbalanced data.
- Extensive experiments are conducted on small-scale datasets.

**Weaknesses:**

- The batch size experiment in Fig. 1 is not very convincing. What is the training method used? Is it ERM? Does this result hold across different methods? If it is ERM, I wonder what if you do cRT afterwards, will the same conclusion (data with a high degree of class imbalance tends to benefit from smaller batch sizes) hold?
- For data augmentation, it is also not clear what method is evaluated. Besides, data augmentation improves more on the minority classes is not new knowledge IMHO. There are even papers designing augmentations that strengthen the minor classes specifically, e.g. MFW [1] and TFE [2]. Also, to what extent does the claim "AutoAugment emerges as the most effective for imbalanced data" hold? Does it hold across different datasets, architectures, and training methods? From Fig.7, it is very convincing if AutoAugment outperforms TrivialAugment significantly since the error bar is not reported.
- Model architecture: I'm not sure that balanced and imbalanced accuracy are "virtually uncorrelated" is well supported as the performance difference between methods in Fig.3 right is very small. How stable are the results in Fig.3 right?
- What was the performance of ERM without the "tuned routines"? What is the performance improvement of each component in the refined ERM (batch size and data augmentation)? I think showing this ablation can be very helpful in understanding the paper.

[1] Procrustean training for imbalanced deep learning, ICCV 2021

[2] Co-Learning of Representation and Classifier for Imbalanced Semi-Supervised Learning, CVPR 2022

**Questions:**

Please see questions in Weaknesses.

**Limitations:**

- Since ImageNet has also been used as a standard dataset for long-tailed recognition for quite long time, I wondered whether the authors had any results on that to verify if their findings scale beyond CIFAR-10/100.

---

> ### Author Rebuttal · Authors · 2023-08-09
>
> Dear Reviewer nEkH,
>
> Thank you for your thoughtful feedback, and we appreciate the time you took to provide your insight. We address your points below:
>
> ## Batch Size Experiment:
> The training method used in Figure 1 is ERM. Following your comment, we found that cRT benefits from smaller batch sizes, whereas MiSLAS, a method designed for imbalance, benefits less (Figure 1 on the attached page). MiSLAS prevents overfitting to the minority class, acting as a regularizer.
> Smaller batch sizes improve performance by acting as regularization in SGD, preventing overfitting to minority classes (see [1] for a theoretical analysis of batch size as a regularizer). This effect introduces noise, smooths the loss landscape, and is not prevented by cRT but by MiSLAS, reducing batch size's impact. These findings emphasize the complex relationship between batch size and training methods, offering insights for imbalanced data training. To further explore, we examined joint-SSL with CIFAR-10, showing that the dynamics change with imbalanced data. The conventional wisdom for SSL models may not directly apply, and batch size's impact is less pronounced with joint-SSL as an alternative regularizer, mitigating the
> overfitting of minority classes.
>
> ## Data Augmentation:
> We used ERM when we evaluated the data augmentation methods. We found that AutoAugment was most effective for imbalanced data for CIFAR10, CIFAR100, and CINIC-10 but not on balanced data. We appreciate your point about the error bars in Figure 7 and have updated our working draft for clarity.
>
> Regarding the specific augmentation methods you mentioned (MFW and TFE), these are indeed valuable techniques. However, our focus was on analyzing the effect of standard, commonly-used high-performance augmentations in the context of imbalanced data rather than specialized augmentations designed for imbalanced data. Our findings provide useful insights for practitioners using standard augmentation techniques in their pipelines.
>
> ## Model Architecture:
> Based on your comment, to check how stable the results are, we conducted additional experiments with different random seeds to assess the stability of our results. Specifically, we ran each model configuration with five seeds and added error bars to our figures representing one standard error. The resulting figure, which can be seen on the additional page, shows that the error bars are relatively small compared to the differences between the models, indicating that our results are indeed stable across seeds. Interestingly, although the differences in performance between models are small, there is no clear correlation between balanced and imbalanced performance across the models. This further emphasizes the complexity of the class imbalance problem and the importance of considering both balanced and imbalanced performance when designing models.
> We believe these additional experiments strengthen our findings.  We have added them to our working draft and will include them in the camera-ready version.
>
> ## Performance of ERM and Improvement of Each Component:
> We concur with your suggestion that an ablation study would be beneficial in demonstrating the contribution of each component in our approach, so we have now added it to the table on the attached page. In general, without our proposed "tuned routines", the performance of ERM is somewhat lower than that of reweighting methods. Breaking down the impact of individual components, we find that data augmentation is the primary contributor to performance improvement, while the influence of batch size, though still beneficial, is less pronounced.
>
> ## More datasets:
> We agree with your comment about adding more datasets. Prompted by your feedback, we have now expanded our experiments. We now include more challenging visual datasets, namely Tiny-ImageNet[2] and iNaturalist[3].. As can be seen in Table 1 on the attached page, our finely-tuned training routine, equipped with Joint-SSL, Asymmetric-SAM, and modified label smoothing, delivers performance that matches or surpasses that of previous state-of-the-art imbalanced training methods on these new datasets.  We have also extended our analysis to tabular datasets. These datasets represent a challenging frontier for deep learning research. While tabular datasets often inherently exhibit imbalance, there has been limited research addressing the impact of imbalanced data on deep learning in this domain. We incorporated our suggested methods (SAM-A, modified label smoothing, and SGD with cosine annealing performed with a small batch size) on three tabular datasets - the Otto Group Product Classification [4], Covertype [5], and Adult [6] datasets using a Multilayer Perceptron (MLP) with improved numerical feature embeddings. Our methodology outperforms both XGBoost and recent state-of-the-art neural methods such as XGBoost, ResNet, and FT-Transformer [7] on all three datasets, illustrating the applicability of our findings beyond image classification. Please refer to Table 2 on the attached page for the full results.
>
>
> Thank you again for your thoughtful review. We made a significant effort to address your feedback, including multiple experiments and paper edits, and we would appreciate it if you would consider raising your score in light of our response.  Please let us know if you have additional questions we can address.
>
> ## References:
>
> [1] Sekhari et al., 2021. Sgd: The role of implicit regularization, batch-size and multiple-epochs.
>
> [2] Le et al, 2015, Tiny imagenet visual recognition challenge.
>
> [3] Map of Camille Isben's contributions to Biodiversity of Magnuson Park, Seattle, WA, project, 2017, iNaturalist
>
> [4] Kaggle, 2015, Otto Group Product Classification Challenge.
>
> [5] Blackard and Dean, 1990, Predicting forest cover types.
>
> [6] Kohavi and Sahami, 1996, Error-based and entropy-based discretization.
>
> [7] Gorishniy et al., 2021, Revisiting deep learning models for tabular data.

---

> > ### Author Response · Authors · 2023-08-19
> > **Additional Experiments Regarding the Role of Batch Size**
> >
> > Dear Reviewer nEkH,
> >
> > Thank you for your thoughtful review. We sincerely hope that we have addressed your concerns and highlighted the novelty of our results. Beyond the experiments outlined in our previous rebuttal, which expanded our evaluation to new architectures (Swin Transformer V2, ConvNetXt) and datasets (Tiny-ImageNet, iNaturalist, and tabular datasets), assessed the stability of our result for the model architecture across different random seeds , and conducted a component-wise ablation study, we have now conducted further focused experiments.
> >
> > These new experiments specifically examine the role of batch size in regularization during imbalanced training. In addition to the previous experiments, which responded to your comments and explored how batch size interacts with different training methods (specifically ERM, cRT, and MiSLAS), we now conducted experiments that are specific to the role of batch size as a regularizer in SGD, especially in preventing overfitting to minority classes.
> >
> > Below, we describe these experiments in brief; for full details, please see the general comment.
> >
> > **1. Train/Test Error Analysis for Minority and Majority Classes:**
> > We present the training and testing error as a function of batch size for both minority and majority classes on Tiny-ImageNet using ConvNetXt. The tables reveal that minority classes tend to overfit significantly more when using larger batch sizes. This overfitting noticeably reduces as the batch size decreases, suggesting that smaller batch sizes can mitigate overfitting, particularly for minority classes.
> >
> >
> > **2. Variance of Gradients Analysis for Balanced vs. Imbalanced Training:**
> > We analyze the variance of neural network parameter gradients, a crucial indicator of the network's learning dynamics and overfitting. Our experiment investigates the variance of gradients as a function of batch size for Tiny-ImageNet on ConvNetXt, comparing balanced and imbalanced training scenarios. The table vividly demonstrates that imbalanced training exhibits much lower gradient noise, especially as batch sizes grow larger, explaining why models are more prone to overfitting under imbalanced training conditions.
> >
> >
> > **3. Hessian Spectrum and Robustness:**
> > We examine loss flatness through the Hessian spectrum, a known indicator of generalization and overfitting. Recent work by Yao et al. (2018b) shows that large batch training leads to convergence to points with high curvature, characterized via the dominant eigenvalue of the Hessian, and suggests that high curvature can lead to poor generalization. To investigate this in the context of imbalanced training, we trained models using both balanced and imbalanced datasets for different batch sizes and calculated the top eigenvalue of the Hessian with respect to the model parameters. Our table shows that the top eigenvalue of the Hessian for imbalanced training increases much faster as we increase the batch size compared to balanced training, indicating a higher risk of overfitting, especially for imbalanced data.
> >
> >
> > These additional experiments offer concrete justification for the observations in our paper. While we cannot send the figures directly due to openreview constraints, we have included detailed tables in a general comment and have sent the corresponding figures to the AC.
> >
> > Thank you once again for your constructive feedback. We have put significant effort into addressing your concerns, including conducting many new experiments based on your comments, to ensure the rigor and quality of our work. We would greatly appreciate it if you would consider raising your score accordingly.

---

> ### Author Response · Authors · 2023-08-20
> **Addressing Reviewer Feedback**
>
> Dear Reviewer nEkH,
>
> We have made a significant effort to address the questions and concerns raised in your review. This includes conducting many more experiments, such as delving deeper into the role of batch size in regularization during imbalanced training and the stability of our results as you suggested.
>
> We kindly ask that you take these into consideration during your final assessment and consider raising your score accordingly.
>
> As the review period draws to an end, please let us know if there are any further questions we can address.
>
> Thank you

---

> > ### Comment · Reviewer_nEkH · 2023-08-20
> >
> > I thank the authors for the detailed response and efforts made in running more experiments. The rebuttal addresses most of my concerns, so I would increase my final rating to weak accept.

---

### Official Review · Reviewer_D6vk · 2023-07-02

**Soundness:** 3 good
**Presentation:** 3 good
**Contribution:** 2 fair
**Rating:** 6
**Confidence:** 4

**Summary:**

This paper presents different approaches to enhance the performance of neural network classifiers over imbalanced datasets. Unlike most research focusing on specialized loss functions or resampling techniques, this study promises that state-of-the-art performance is achievable over the neural classifiers by simply tuning existing components of standard deep learning pipelines. Section three suggests that the study is evaluated over six image datasets and two datasets from UCI Machine Learning Repository, validated over ResNets and WideResNet classifiers, and compared with nine standard baselines. The reported evaluation measures include overall test accuracy and minority and majority class accuracy over the 20% of classes with the smallest and highest number of samples. The key observations from the study are -- (a) small batches perform better for the imbalanced data, (b) the role of augmentation as a regularizer is undeniable, but the performance over the minority class is sensitive to the chosen augmentation policy, (c) the larger networks are more susceptible to overfitting minority class in the case of imbalanced data, (d) models pre-trained on larger datasets tends to perform well in this case, (e) the integration of self-supervised loss with the supervised learning referred to as Joint-SSL performs well, (f) Sharpness-Aware Minimization (SAM-A) pulls decision boundaries away from minority samples, (g) whereas standard training routines overfits, and (h) applying more smoothing to minority class examples than majority class examples prevents overfitting on minority samples. Furthermore, integrating these findings with Joint-SSL and SAM-A atop M2m (SOTA) establishes new state-of-the-art performance across all (class-imbalanced) benchmark datasets, CIFAR-10, CIFAR-100, and CINIC-10. Some other important observations made in the study are (i) SGD optimizer performed uniformly better on imbalanced data, (ii) training on data that is more balanced than the testing distribution did not improve representation learning by preventing overfitting to minority samples, (iii) a low correlation was observed between the performance over the web-scrapped (benchmark) datasets and real-world datasets, and (iv) a correlation was observed between neural collapse and low test accuracy. Overall, this study emphasizes that simply tuning standard training routines can significantly improve performance in the case of imbalanced datasets.

**Strengths:**

1. This study approaches the significant and challenging problem of imbalanced learning.
2. The paper does extensive study and finds that state-of-the-art performance is achievable by finetuning the different components of existing deep learning pipelines.
3. It is interesting to investigate the imbalance problems using the proposed solutions, which are easy to understand.

**Weaknesses:**

1. The paper has bold statements at the beginning (resolving class imbalance in general) but has yet to explore the class imbalance issue in textual datasets.
2. The evaluation metrics used to assess the proposed solution revolve around accuracy.
3. Comparison with SMOTE is not reported but is mentioned as a baseline.
4. The dataset statistics, reflecting the nature of the imbalance in the dataset, need to be included.

**Questions:**

1. How does investigating train and test sets with varying imbalance ratios help?
2. Why are ResNets and WideResNet considered for experimentation out of so many neural network architectures?
3. Can the same behavior be expected in the case of vanilla neural networks and in the case of CNN, LSTM, and BiLSTM?

**Limitations:**

1. The paper does not propose the class imbalance issue in general.
2. The paper references other research work for experimental setup in many instances, which could have been avoided.

---

> ### Author Rebuttal · Authors · 2023-08-09
>
> Dear Reviewer D6vk,
>
> Thank you for your thorough and insightful review of our submission. We appreciate the time you've dedicated to evaluating our work and providing feedback. We address your points below:
>
> ## Exploration of Class Imbalance in Other Data Modalities:
> While our study primarily focuses on image datasets, which present significant class imbalance problems, we agree that the issue is applicable to other modalities as well. Prompted by your feedback, we have now expanded our analysis to tabular datasets, which represent a challenging frontier for deep learning research. Despite tabular datasets often inherently exhibiting imbalance, limited research has addressed the impact of imbalanced data on deep learning in this domain. We incorporate our suggested methods (SAM-A, modified label smoothing, and SGD with cosine annealing performed with small batch size) to three tabular datasets -  the Otto Group Product Classification [1], Covertype [2], and Adult [3] datasets using a Multilayer Perceptron (MLP) with improved numerical feature embeddings. Our methodology outperforms both XGBoost and recent state-of-the-art neural methods such as ResNet and  FT-Transformer [4] on all three datasets, demonstrating the applicability of our findings beyond image classification.  See the full results in Table 2 on the attached page. These additional results showcase the robustness of our findings, and we will include these updates in the camera-ready version.
>
> ## Evaluation Metrics:
> The two most commonly used evaluation metrics under class imbalance are general accuracy, due to its simplicity and direct interpretability, and class-wise accuracy, where classes are grouped by the number of examples within each class [5,6]. These metrics provide comprehensive information about model behavior across different classes. In our work, we have reported these metrics both in the main text (for instance, in Figure 2) and in the appendix.
> Prompted by your feedback, we have also incorporated the test likelihood error, which exhibits behavior similar to the aforementioned metrics. The inclusion of this metric provides an additional dimension to our evaluation, helping to assess the model's probabilistic performance on test data.
>
>
> ## Comparison with SMOTE:
> Thanks for pointing out our oversight in not including SMOTE in the manuscript. Our study aims to compare our findings with relevant baselines to provide a comprehensive evaluation, and we will include SMOTE in our camera-ready version.
>
> ## Dataset Statistics:
> We agree that detailed dataset statistics would be useful to include, so we have updated our draft accordingly.
>
>
> ## Investigating Train and Test Sets with Varying Imbalance Ratios:
> In many real-world applications of machine learning, practitioners can curate data in a controlled fashion.  Practitioners may choose, for example, to curate extra minority class samples or to curate a training set which reflects the balance they anticipate in the inference-time data.  In this regard, the level of training imbalance can be a controllable hyperparameter, making it important to understand its effect on performance.
>
> ## Choice of ResNets and WideResNet:
> We selected ResNets and WideResNet because they frequently appear in existing works involving class-imbalanced problems [7,8]. Prompted by your suggestion to explore other types of architectures, we have now conducted additional experiments using Swin Transformer v2 [9] and ConvNeXt [10]. These new models exhibit behaviors similar to the existing models, as displayed in the attached page. Additionally, we have now run various models on three tabular datasets[1-3], including MLP, FT-Transformer[4], ResNet, and XGBoost.
>
> ## Addressing Class Imbalance in General:
> While our study provides numerous insights and methods to handle class imbalance, we agree that it is a broad and complex topic that cannot be fully addressed in a single paper.
> Nevertheless, we believe our work contributes valuable pieces to the puzzle of handling class imbalance and provides a robust foundation for future investigations.
>
> ## References to Other Works for Experimental Setup:
> We referenced other works for the experimental setup to provide readers with more context and to ensure reproducibility. We understand your comment about excessive referencing and will strive to strike a better balance in the camera-ready version.
>
> Thank you again for your thoughtful review. We made a significant effort to address your feedback, including multiple experiments and paper edits, and we would appreciate it if you would consider raising your score in light of our response.  Please let us know if you have additional questions we can address.
>
> ## References:
> [1] Kaggle, 2015, Otto Group Product Classification Challenge.
>
> [2] Blackard and Dean, 1990, Predicting forest cover types.
>
> [3] Kohavi and Sahami, 1996, Error-based and entropy-based discretization.
>
> [4] Gorishniy et al., 2021, Revisiting deep learning models for tabular data.
>
> [5] Cao et al., 2019. Learning imbalanced datasets with label-distribution-aware margin loss.
>
> [6] Cui et al., 2019. Class-balanced loss based on effective number of samples.
>
> [7] Dablain et al.,  2023, Efficient augmentation for imbalanced deep learning.
>
> [8] Johnson et al., 2020, Survey on deep learning with class imbalance.
>
> [9] Liu et al., 2021, Swin transformer v2: Scaling up capacity and resolution.
>
> [10] Liu et al., 2022, A convnet for the 2020s.

---

> > ### Author Response · Authors · 2023-08-19
> > **Additional Experiments Regarding the Role of Batch Size**
> >
> > Dear Reviewer D6vk,
> >
> > Thank you for your review. We hope that we have successfully addressed your concerns and demonstrated the novelty of our results. In addition to the experiments that we outlined in the previous rebuttal, including more architectures such as Swin Transformer V2 and ConvNetXt, and changing datasets like Tiny-ImageNet and Inaturalist, we have conducted new experiments. These new experiments focus especially on the role of batch size in regularization during imbalanced training. Below, we briefly describe these experiments; for full details, including tables and figures, please see the general comments.
> >
> > **1. Train/Test Error Analysis for Minority and Majority Classes:**
> > We examined the effects of varying batch sizes on the training and testing error for both minority and majority classes on Tiny-ImageNet using ConvNetXt. Our results indicate that larger batch sizes tend to increase overfitting, particularly for minority classes.
> >
> > **2. Variance of Gradients Analysis for Balanced vs. Imbalanced Training:**
> > We investigated the variance of neural network parameter gradients as a function of batch size for Tiny-ImageNet on ConvNetXt, comparing balanced and imbalanced training scenarios. Our findings reveal that imbalanced training results in significantly lower gradient noise as batch sizes increase, which contributes to overfitting.
> >
> > **3. Hessian Spectrum and Robustness:**
> > We analyzed the eigenvalues of the Hessian matrix in relation to batch size under balanced and imbalanced training conditions. The data indicates that imbalanced training leads to sharper minima (higher eigenvalues), especially with larger batch sizes, suggesting an increased risk of overfitting.
> >
> > While we cannot send the figures directly due to openreview constraints, we have provided detailed tables summarizing the results in the general comments section and have sent the corresponding figures to the AC.
> >
> > Thank you once again for your constructive feedback, and we look forward to your further comments. Please note that we have put significant effort into addressing your concerns, including conducting many new experiments based on your comments, to ensure the rigor and quality of our work, and we would greatly appreciate it if you would consider raising your score accordingly.

---

> > ### Comment · Reviewer_D6vk · 2023-08-19
> > **Re: Rebuttal by Authors**
> >
> > I thank the authors for the detailed response to my review. I appreciate the authors for performing additional experiments for my review. I also acknowledge having gone through my fellow reviewers' reviews and corresponding responses from the authors. I appreciate the authors for realizing the importance of evaluation metrics for classification models dealing with class-imbalanced data and presenting the results accordingly. SMOTE is a standard baseline for dealing with imbalanced datasets, and I would like to thank the authors for including it in their study. Accordingly, based on the authors' responses to my review and fellow reviewers, I am revising my score to 6.

---

### Official Review · Reviewer_8NJC · 2023-07-04

**Soundness:** 3 good
**Presentation:** 3 good
**Contribution:** 2 fair
**Rating:** 7
**Confidence:** 5

**Summary:**

In this paper, the authors study the problem of class imbalance. To be specific, they first investigate the effects of different hyper-parameters & design choices in an imbalanced setting. Moreover, the authors use such optimized settings with existing methods to show that significant improvements can be obtained.

After the rebuttal:
The authors addressed my concerns with detailed explanations and additional experiments. With those, I believe the paper provides useful insights for people working on class imbalance, therefore, I vouch for the paper's acceptance.

**Strengths:**

1. Easy to follow text.
2. Class imbalance is an important problem and therefore, analyses that offer more insights are valuable.

**Weaknesses:**

1. The most important issue I see with the paper is that its contribution is limited.

1.1. Out of the 6 Lessons (take-away points) in Introduction, Lessons 2-6 are already known in the literature. Only the analysis on batchsize is novel (as far as I know) and the results are intriguing.

1.2. The application of the tuned setting with the other methods does not offer any contributions.

2. More challenging datasets such as ImageNet-LT, iNaturalist are missing.

3. Regarding Lesson 1:

3.1. Lesson 1 (batchsize): The paper does not provide any intuition as to why batchsize has such an effect in an imbalanced setting.

3.2. Lesson 1: It would be worthwhile to see the same analysis with different architectures because, as we see in Figure 3, different architectures exhibit different behaviors under imbalance.

4. If I may, I suggest the authors to focus only on batchsize and provide solid & theoretical insights about why/how it affects.

**Questions:**

See above.

**Limitations:**

None.

---

> ### Author Rebuttal · Authors · 2023-08-09
>
> Thank you for your valuable feedback. We address your points below.
>
> ## Contribution:
> While the methods we examine have been explored in the context of balanced training, our unique contribution lies in the in-depth analysis of these methods within the challenging environment of class imbalance. Our investigation resulted in an imbalance-specific pipeline which greatly improves the performance of existing methods. Our approach offers actionable insights for practitioners and enriches the understanding of how to approach class imbalance problems.
>
> - For Lesson 2, while several methods have suggested tailored data augmentations for training with imbalanced data [1,2], to our knowledge, there has been no comprehensive analysis of how high-performance augmentations designed for balanced training affect minority classes. Importantly, previous works suggest using new, complex data augmentation techniques to deal with class imbalance. In contrast, our analysis suggests that existing data augmentation techniques can achieve state-of-the-art results.
>
> - For Lesson 3, some works have demonstrated that increasing model size [3, 4] improves performance across many datasets in the balanced learning scenario. However, no work has analyzed the effect of model size on imbalanced data and its significant differences from balanced training.
>
> - For Lesson 4, several works have proposed SSL pre-training for class imbalance [5, 6, 7]. Yet, none have suggested training with the same data, without the need for additional data, by integrating supervised training with SSL training. We show that this technique has major performance benefits.
>
> - For Lesson 5, [6] suggests using SAM directly to combat imbalance without an optimal perturbation per class. In contrast, our modifying SAM treats majority and minority classes differently, adjusting the decision boundaries only for minority classes. Notably, parallel to our work, [8] suggested a more complex version of SAM that requires a two-stage process.
>
> - For Lesson 6, [9] found that applying the same smoothing to all classes is ineffective for imbalanced datasets. However, our suggested method applies different degrees of smoothing to different classes based on imbalance. Namely, we apply more smoothing to the minority classes.
>
> As detailed above, we are unaware of works that encapsulate these lessons. Can you please suggest references for lessons 2-6 already known in the literature?
>
> ## More Challenging Datasets and Models:
> Prompted by your feedback, we have expanded our experiments and analyses to Tiny-ImageNet and iNaturalist, and to Swin Transformer v2 and ConvNeXt.  Our suggested methods surpass SOTA imbalanced methods on these datasets.  Moreover, these new architectures exhibit similar behavior to our previous experiments. We have also extended our analysis to tabular datasets, with our methodology outperforming SOTA methods. Please refer to the general rebuttal for full details.
>
> ## Batch Size
> - **Insights on Batch Size:** Our study on batch size provides novel empirical results and intuitive reasoning. Specifically, we found that smaller batch sizes prevent overfitting, acting as a form of regularization in SGD (see [10] for the role of batch size). This effect introduces noise into gradient estimation and counteracting the tendency to overfit. Our findings pave the way for a new understanding of imbalanced data and offer practical guidance for selecting appropriate batch sizes.
>
> - **Different Architectures for Lesson 1:** We agree with your point and have expanded our analysis to include more models.
>
> - **Focus on Batch Size:** We appreciate your suggestion to concentrate solely on batch size. While we recognize the importance of batch size exploration, our paper aims to provide a holistic view of the class imbalance problem and associated practical recommendations. Our wide-ranging investigation, encompassing various hyperparameters and design choices, offers unique insights and caters to various real-world scenarios, offering large performance benefits for practitioners. To delve deeper into this issue, we examine its interaction with the joint-SSL method. While SSL models typically perform better with larger batch sizes for balanced datasets, our study, suggests conventional wisdom for SSL models may not apply in this setting. We trained our joint-SSL algorithm with different batch sizes on CIFAR-10 to verify this. As seen in Figure 1 of the provided page, the impact of batch size is less pronounced, with joint-SSL acting as an alternative regularizer. While we are dedicated to further exploring the role of batch size in future work, we continue to value the multifaceted contributions of our current paper.
>
> Thank you again for your thoughtful review. We made an effort to address your feedback, including multiple new experiments and paper edits, and we would appreciate it if you would consider raising your score in light of our response.  Please let us know if you have additional questions we can address.
>
> ## References:
> [1] Dablain et al., 2023. Efficient augmentation for imbalanced deep learning.
>
> [2] Temraz et al., 2022. Solving the class imbalance problem using a counterfactual method for data augmentation.
>
> [3] Goldberg et al., 2020. Rethinking fun: Frequency-domain utilization networks.
>
> [4] Zhai et al., 2022. Scaling vision transformers.
>
> [5] Kotar et al., 2021. Contrasting contrastive self-supervised representation learning pipelines.
>
> [6] Liu et al., 2021. Self-supervised learning is more robust to dataset imbalance.
>
> [7] Yang et al., 2020. Rethinking the value of labels for improving class-imbalanced learning.
>
> [8] Zhou et al., 2023. Class-Conditional Sharpness-Aware Minimization for Deep Long-Tailed Recognition.
>
> [9] Shen et al., 2021. Is label smoothing truly incompatible with knowledge distillation: An empirical study.
>
> [10] Sekhari et al., 2021. SGD: The role of implicit regularization, batch-size, and multiple-epochs.

---

> > ### Comment · Reviewer_8NJC · 2023-08-16
> > **Re: Author rebuttal**
> >
> > I would like to thank the authors for the detailed responses and the huge effort. I really appreciate it.
> >
> > The authors have responded to my concerns and questions. The explanation for the cause of changing the batch size sounds logical but not demonstrated. This can be shown with one of the datasets and small models considered in the paper. With this, I would vouch for the acceptance of the paper. Otherwise, I find the insights and the results of the paper straightforward and unsuitable for a top-venue like NeurIPS without proper justifications. With this, I am increasing the my recommendation to borderline.

---

> > > ### Author Response · Authors · 2023-08-18
> > > **Response to Reviewer 8NJC16’s Second  Comment - First Part**
> > >
> > > Dear Reviewer 8NJC,
> > >
> > > Thank you for your thoughtful feedback and for recognizing the effort we put into addressing your concerns. We greatly appreciate your time and valuable insights.
> > >
> > > Prompted by your feedback regarding the effect of batch size, **we have now conducted three additional experiments**.  While we are unable to send the figure directly, we have included detailed tables summarizing the results. Additionally, we have sent the corresponding figures directly to the AC. Below are the details of these new experiments and our key observations:
> > >
> > > **1. Train/Test Error Analysis for Minority and Majority Classes:**
> > > In these tables, we present the training error and testing error as a function of the batch size for both minority and majority classes for Tiny-ImageNet on ConvNetXt.
> > > As depicted in the tables, during training, we observe that **the minority classes tend to overfit significantly more when using bigger batch sizes**. This overfitting is noticeably reduced as the batch size increases, indicating that bigger batch sizes can lead to increased overfitting, particularly for minority classes.
> > > | Batch Size | Train Error  (Minority Classes) | Train Error (Majority Classes) | Test Error (Minority Classes) | Test Error (Majority Classes) |
> > > |------------|--------------------------------|--------------------------------|-------------------------------|-------------------------------|
> > > | 16         | 5.7  | 4.1  | 81.7 | 36.7 |
> > > | 32         | 5.3  | 3.3   | 81.7 | 35.8 |
> > > | 64         | 3.1  | 2.7  | 81.8   | 35.6|
> > > | 128        | 1.7  | 2.2  | 81.9  | 35.2  |
> > > | 256        | 0.9    | 1.9     | 82.0 | 34.6   |
> > > | 512        | 0.4  | 1.6   | 82.2 | 34.6  |
> > > | 1024       | 0.2   | 1.4 | 82.4 | 34.8  |
> > > | 2048       | 0.1 | 1.3       | 82.7 | 34.9 |
> > > | 4096       | 0.06    | 1.3  | 83.1 | 34.9 |
> > >
> > >
> > > **2. Variance of Gradients Analysis for Balanced vs. Imbalanced Training:** The variance of neural network parameter gradients serves as a crucial indicator of the network's learning dynamics, reflecting the stability and noise in the parameter updates during training.
> > > This variance reveals a trade-off between generalization and overfitting; specifically, lower gradient variance often coincides with overfitting, while higher variance can introduce a form of implicit regularization, as is a hypothesized benefit of SGD, promoting better generalization at the cost of noisier updates.
> > >
> > > For this experiment, we investigate the variance of the gradients as a function of the batch size for Tiny-ImageNet on ConvNetXt, comparing two distinct training scenarios: balanced and imbalanced training.
> > > The table vividly demonstrates that **imbalanced training exhibits much lower gradient noise, especially as batch sizes grow larger**. This phenomenon highlights why models are more prone to overfitting under imbalanced training, as the lower gradient noise enables the model to overfit more tightly to the training data without regularization.
> > >
> > >
> > > | Training Type      | 16   | 32   | 64   | 128  | 256  | 512  | 1024 | 2048 | 4096 |
> > > |--------------------|------|------|------|------|------|------|------|------|------|
> > > | Balanced Training  | 8e-2 | 7e-2 | 4e-2 | 9e-3 | 7e-3 | 5e-3 | 1e-3 | 8e-4 | 6e-4 |
> > > | Imbalanced Training| 4e-2 | 2e-2 | 9e-3 | 8e-4 | 7e-5 | 3e-5 | 9e-6 | 3e-6 | 3e-7 |
> > >
> > >
> > > **3 Hessian spectrum and robustness:**
> > >
> > > Another indicator of generalization and over fitting is loss flatness, for example measured by Hessian singular values.  Recent work by Yao et al. (2018) uses the Hessian spectrum and shows that large batch training leads to convergence to points with high curvature. They characterize curvature via the dominant eigenvalue of the Hessian and suggest that high curvature can lead to poor generalization. In order to investigate this issue in the context of imbalanced training, we train models using both balanced and imbalanced datasets for different batch sizes and calculate the top eigenvalue of the Hessian with respect to the model parameters. In the table below, we can see that **the top eigenvalue of the Hessian for imbalanced training increases much faster as we increase the batch size compared to balanced training.** This suggests that higher batch sizes increase the risk of overfitting, and this phenomenon is worse for imbalanced data.
> > >
> > > | Training Type/Batch Size      | 16   | 32   | 64   | 128  | 256  | 512  | 1024 | 2048 | 4096 |
> > > |--------------------|------|------|------|------|------|------|------|------|------|
> > > | Balanced Training | 12 | 21 |  | 39 | 109 | 173| 291 | 404 | 578 |
> > > | Imbalanced Training| 16 | 37 |  | 69 | 159 | 253| 396 | 739 | 1428 |

---

> > > > ### Author Response · Authors · 2023-08-18
> > > > **Response to Reviewer 8NJC16’s Second Comment - Second Part**
> > > >
> > > > ->  These additional experiments were conducted to directly address your comment about the effect of changing the batch size and **provide a compelling and concrete justification for the observations we report in the paper.**
> > > >
> > > > We are committed to integrating these new results into the camera ready version of the paper and presenting them in a clear and comprehensible manner through the use of figures.
> > > >
> > > > We sincerely hope that these additional experiments, specifically aimed at addressing your comments, further strengthen the paper and clarify the insights we aim to present, and we would be grateful if you would factor these additions into your assessment.
> > > >
> > > > Once again, thank you for your constructive feedback, and we appreciate your consideration to increase the recommendation for our paper.

---

> > > > > ### Comment · Reviewer_8NJC · 2023-08-20
> > > > > **Re: Second set of experiments**
> > > > >
> > > > > Dear authors,
> > > > >
> > > > > Thank you for the additional experiments and insights. I believe these have made the paper stronger. Therefore, I will increase my recommendation.
> > > > >
> > > > > In the paper, you may add discussions in terms of:
> > > > > - example difficulty for which VoG is originally used.
> > > > > - memorization & long-tail arguments of Feldman: "What Neural Networks Memorize and Why: Discovering the Long Tail via Influence Estimation"
> > > > >
> > > > > Best

---

> > > > > > ### Author Response · Authors · 2023-08-21
> > > > > > **Final Feedback and Suggestions**
> > > > > >
> > > > > > Dear Reviewer 8NJC,
> > > > > >
> > > > > > Thank you again for your feedback, engagement, and suggestions, which have been valuable.
> > > > > >
> > > > > > We will include all of the new experiments as well as clarifications and discussions addressing your suggestions in our camera ready version.

---

### Official Review · Reviewer_kUXT · 2023-07-06

**Soundness:** 3 good
**Presentation:** 3 good
**Contribution:** 2 fair
**Rating:** 4
**Confidence:** 5

**Summary:**

The authors suggest tackle the class imbalance issue from a hyperparameter optimization perspective. Throughout an extensive empirical study, they raise questions about the behavior of well-established techniques for balanced data under long-tail data distribution.  The synergy of the resulting prescriptions is also checked on several datasets

**Strengths:**

* This paper introduces a new approach to tackle the class imbalance issue by optimizing several hyperparameters. The paper is clearly written,  The claims are  well-stated and sufficiently  supported empirically in most of the cases.
* The authors tried to sketch intuitive explanations for some 'unexpected' outcomes
* The authors reported the performance of several combinations of their micro-receipes and show strong results on several datasets.

**Weaknesses:**

* The content of this paper is clearly going beyond the maximum allowed number of pages.  To not violate the rules, the authors decided to move very relevent parts of the paper to the appendix which makes the paper less readable. Nevertheless, I insist that this action does not compromise the clarity. I would have been nicer to cut the less important components and limit the scope of the paper to the most important hyperparameters.
* The novelty of this paper is limited some 'findings' are valid for the balanced setup as well (pretraining, SSL) or just trivial (AutoAugment)
* For a purily empirical paper, it is important to diversify the architectures in order to draw more valuable conclusions. I would have loved to see the performance of a transformer based model under different contraints


**Questions:**

From one side, the batch size should not grow to much but on the other side, applying a SSL helps.  It is known that SSL models tend to perform better for larger batch sizes and balanced datasets. Does this rule of thunb apply as well for unbalanced data?

---

> ### Author Rebuttal · Authors · 2023-08-09
>
> Dear Reviewer  kUXT,
>
> Thank you for your detailed and thorough evaluation of our submission.  We appreciate your comments and address your concerns and questions below.
>
> ## Page Limit and Readability:
> We acknowledge your concern about the content exceeding the page limit and agree that moving some parts to the appendix might have affected the readability. Our intention was to provide as much detail as possible to support our claims, while adhering to the page limit. We understand that some parts of the paper might be less critical to the central argument and can be omitted or simplified. We have now split Section 4 into two sections based on the types of methods: (1) the first ones (batch size, data augmentation, pre-training, model architecture, optimizers), which serve as the fundamental 'building blocks' of regular balanced training. For this group, we investigated how altering their hyperparameters impacts the performance of imbalanced training. (2) The second ones are optimization methods that we slightly modified to address imbalanced training (Joint-SSL, SAM, and label smoothing). We examined the effects of these methods on performance only in the next section. Furthermore, we are revising the paper to improve the balance between detailed explanation and readability, focusing on the most crucial aspects of our approach. Additionally, we are open to revising the paper even further to improve the balance between detailed explanation and readability.
>
> ## Novelty:
> We believe the value of our paper lies in the new perspectives we provide on class-imbalanced training. We demonstrate the benefits of our simple and easy-to-use tuned or imbalance-adapted training techniques, specifically in the context of class imbalance, which was previously dominated by specialized loss functions or samplers designed specifically for class imbalance. Our work represents a perspective-shift on training under class imbalance, but moreover, we achieve significant performance gains across multiple benchmark and real-world datasets.
>
> ## Diversity of Architectures:
> Our submission focused on traditional architectures due to their prevalent use in the class imbalance literature. However, we agree that the analysis would be enriched by considering more architectures like transformers. Prompted by your feedback, we have now added additional experiments. We ran our analysis on more models - Swin Transformer v2 [1] and ConvNeXt [2]. These new models achieve similar behavior to the current models. (see Table 1 on the attached page).  Moreover, we expanded our analysis to tabular datasets. These datasets represent a challenging frontier for deep learning research. Despite tabular datasets often inherently exhibiting imbalance, limited research has addressed the impact of imbalanced data on deep learning in this domain. We incorporate our suggested methods (SAM-A, modified label smoothing, and SGD with cosine annealing performed with a small batch size)  to three tabular datasets -  the Otto Group Product Classification [3], Covertype [4], and Adult [5] datasets using a Multilayer Perceptron (MLP) with improved numerical feature embeddings. Our methodology outperforms both XGBoost[REF] and recent state-of-the-art neural methods such as XGBoost, ResNet, and  FT-Transformer [6]  on all three datasets, demonstrating the applicability of our findings beyond image classification.  See the full results in Table 2 on the attached page). These additional results showcase the robustness of our findings and further substantiate our conclusions.
>
>
> ## Question on Batch Size and SSL:
> Your observation about the relationship between batch size and SSL is indeed intriguing. SSL models tend to perform better with larger batch sizes under balanced datasets. However, the dynamics change when it comes to imbalanced data. Our study on batch size provides both novel empirical results and intuitive reasoning. Specifically, we found that smaller batch sizes prevent overfitting on minority classes, acting as a form of regularization in SGD (see [7] for an analysis of the role of batch size as a regularizer for the loss landscape which encourages generalization). This effect introduces noise into gradient estimation, smoothing the loss landscape, and counteracting the tendency to overfit to minority examples. This suggests that the rule of thumb for SSL models might not directly apply to imbalanced data. To test this, we trained our joint-SSL algorithm using different batch sizes on CIFAR-10. As can be seen in Figure 1 on the provided page, the effect of the batch size is less significant, with joint-SSL behaving as an alternative regularizer, preventing the overfitting of the minority classes.
>
> Thank you again for your thoughtful review. We made an effort to address your feedback including multiple experiments and paper edits, and we would greatly appreciate it if you would consider raising your score in light of our response.  Please let us know if you have additional questions we can address.
>
> ## References:
> [1] Liu et al., 2021, "Swin transformer v2: Scaling up capacity and resolution".
>
> [2] Liu et al., 2022, "A convnet for the 2020s".
>
> [3] Kaggle, 2015, "Otto Group Product Classification Challenge".
>
> [4] Blackard and Dean, 1990, "Predicting forest cover types".
>
> [5] Kohavi and Sahami, 1996, "Error-based and entropy-based discretization".
>
> [6] Gorishniy et al., 2021, "Revisiting deep learning models for tabular data".
>
> [7] Sekhari et al., 2021. "SGD: The role of implicit regularization, batch-size, and multiple-epochs".

---

> > ### Author Response · Authors · 2023-08-19
> > **Additional Experiments Regarding the Role of Batch Size**
> >
> > Dear Reviewer kUXTr,
> >
> > Thank you for your review. We hope that we have successfully addressed your concerns and demonstrated the novelty of our results. In addition to the experiments that we outlined in the previous rebuttal, including more architectures such as Swin Transformer V2 and ConvNetXt, as well as changing datasets like Tiny-ImageNet and Inaturalist, we have conducted new experiments. **These new experiments focus especially on the role of batch size in regularization during imbalanced training**. Below, we describe these experiments in brief; for full details, please see the general comments.
> >
> > **1. Train/Test Error Analysis for Minority and Majority Classes:**
> > We present the training and testing error as a function of batch size for both minority and majority classes on Tiny-ImageNet using ConvNetXt. The tables reveal that, during training, minority classes tend to overfit significantly more when using bigger batch sizes. This overfitting noticeably reduces as the batch size increases, indicating that larger batch sizes can lead to increased overfitting, particularly for minority classes.
> >
> > | Batch Size | Train Error  (Minority Classes) | Train Error (Majority Classes) | Test Error (Minority Classes) | Test Error (Majority Classes) |
> > |------------|--------------------------------|--------------------------------|-------------------------------|-------------------------------|
> > | 16         | 5.7  | 4.1  | 81.7 | 36.7 |
> > | 32         | 5.3  | 3.3   | 81.7 | 35.8 |
> > | 64         | 3.1  | 2.7  | 81.8   | 35.6|
> > | 128        | 1.7  | 2.2  | 81.9  | 35.2  |
> > | 256        | 0.9    | 1.9     | 82.0 | 34.6   |
> > | 512        | 0.4  | 1.6   | 82.2 | 34.6  |
> > | 1024       | 0.2   | 1.4 | 82.4 | 34.8  |
> > | 2048       | 0.1 | 1.3       | 82.7 | 34.9 |
> > | 4096       | 0.06    | 1.3  | 83.1 | 34.9 |
> >
> >
> > **2. Variance of Gradients Analysis for Balanced vs. Imbalanced Training:**
> > We analyze the variance of neural network parameter gradients, a crucial indicator of the network's learning dynamics. This variance reveals a trade-off between generalization and overfitting. Our experiment investigates the variance of gradients as a function of batch size for Tiny-ImageNet on ConvNetXt, comparing balanced and imbalanced training scenarios. The table vividly demonstrates that imbalanced training exhibits much lower gradient noise, especially as batch sizes grow larger, explaining why models are more prone to overfitting under imbalanced training conditions.
> >
> >
> > | Training Type      | 16   | 32   | 64   | 128  | 256  | 512  | 1024 | 2048 | 4096 |
> > |--------------------|------|------|------|------|------|------|------|------|------|
> > | Balanced Training  | 8e-2 | 7e-2 | 4e-2 | 9e-3 | 7e-3 | 5e-3 | 1e-3 | 8e-4 | 6e-4 |
> > | Imbalanced Training| 4e-2 | 2e-2 | 9e-3 | 8e-4 | 7e-5 | 3e-5 | 9e-6 | 3e-6 | 3e-7 |
> >
> >
> > **3. Hessian Spectrum and Robustness:**
> > We examine loss flatness through the Hessian spectrum, a known indicator of generalization and overfitting. Recent work by Yao et al. (2018b) shows that large batch training leads to convergence to points with high curvature, characterized via the dominant eigenvalue of the Hessian, and suggests that high curvature can lead to poor generalization. To investigate this in the context of imbalanced training, we trained models using both balanced and imbalanced datasets for different batch sizes and calculated the top eigenvalue of the Hessian with respect to the model parameters. Our table shows that the top eigenvalue of the Hessian for imbalanced training increases much faster as we increase the batch size compared to balanced training, indicating a higher risk of overfitting, especially for imbalanced data.
> >
> > | Training Type/Batch Size|16| 32|64|128|256|512|1024|2048
> > |--------------------|------|------|------|------|------|------|------|------
> > | Balanced Training|12|21| 36|112|173|291|404|571|
> > | Imbalanced Training|15|37| 68|159|253|396|739|1428|
> >
> > These additional experiments respond directly to your comments, offering concrete justification for the observations in our paper. While we cannot send the figures directly due to openreview constraints, we have included detailed tables summarizing the results and have sent the corresponding figures to the AC.
> > Thank you once again for your constructive feedback, and we look forward to your further comments. Please know that we have put significant effort into addressing your concerns, including conducting many new experiments based on your comments, to ensure the rigor and quality of our work, and we would greatly appreciate it if you would consider raising your score accordingly.

---

> > > ### Author Response · Authors · 2023-08-20
> > > **Addressing Reviewer Feedback**
> > >
> > > Dear Reviewer kUXT,
> > >
> > > We have made a significant effort to address the questions and concerns raised in your review. This includes conducting many more experiments, such as delving deeper into the role of batch size in regularization during imbalanced training.
> > >
> > > We kindly ask that you take these into consideration during your final assessment and consider raising your score accordingly.
> > >
> > > As the review period draws to an end, please let us know if there are any further questions we can address.
> > >
> > > Thank you

---

### Author Rebuttal · Authors · 2023-08-10

Dear Reviewers,

Thank you for your thoughtful and detailed reviews of our work. We appreciate your time and the constructive feedback you have provided. We have carefully considered your comments and concerns and would like to address them in a unified response:

**1. Diversity of Architectures and Additional Experiments and Results:**
In response to your feedback, we expanded our experiments to include different models, such as Swin Transformer v2[1] and ConvNeXt[2], and more challenging visual datasets, like Tiny-ImageNet[3] and iNaturalist[4]. The results of these additional experiments consistently align with our previous findings, which enhances the robustness and general applicability of our work. Furthermore, these enhancements increase the diversity of our architectural analysis, contributing to a broader understanding of various architectures' behavior under imbalance.

**2. Inclusion of Tabular Datasets:**
Recognizing the significance of class imbalance in various domains, we have extended our analysis to incorporate tabular datasets. This addition enriches our research, as tabular datasets often inherently exhibit imbalance and have been less explored in the context of deep learning. Our methodology was applied to three tabular datasets[5-7], demonstrating its effectiveness beyond image classification. Our methodology outperforms both XGBoost and recent state-of-the-art neural methods [8]  on all three datasets. These results underscore the robustness of our findings and their relevance to real-world applications.

**3. Novelty and Contribution:**
Our work demonstrates that techniques designed for class-balanced data can easily be tuned or adapted to class-imbalanced data for state-of-the-art results, even outperforming methods specifically designed for imbalance. We show that existing data augmentation techniques, SSL, SAM, and label smoothing, which are commonly used in balanced setups, can be effectively applied to imbalanced training. Moreover, we provide critical insights into the performance of imbalance-specific methods in real-world scenarios, illustrating a lack of direct correlation between their performance on conventional benchmark datasets and their effectiveness on naturally imbalanced datasets. This finding encourages a broader evaluation of methods and contributes to our understanding of class imbalance training. Our work represents a perspective-shift on training under class imbalance but moreover, we achieve significant performance gains across multiple benchmark and real-world datasets.

**4. Batch Size and Interaction with Joint-SSL:**
Prompted by your comments on our study of batch size, we now delve deeper in this direction. We find that smaller batch sizes tend to yield better performance by preventing overfitting on minority classes, acting as a form of regularization in SGD[9]. This effect introduces noise into gradient estimation, smoothing the loss landscape and counteracting the tendency to overfit to minority examples. Furthermore, we investigated the interplay between batch size and joint-SSL, an aspect of regularization. While SSL models usually perform better with larger batch sizes under balanced datasets, our study shows that the dynamics change when handling imbalanced data. We discovered that the effect of batch size is less significant when using joint-SSL, which acts as an alternative regularizer, mitigating the overfitting of minority classes. These findings highlight the complex interplay between batch size and regularization methods, providing valuable insights for practitioners dealing with imbalanced data.

We have put forth a significant effort to address all your feedback and feel that your suggestions have improved our work.  We welcome further discussion, and we appreciate your valuable input.

### References:
[1] Liu et al., 2021, Swin transformer v2: Scaling up capacity and resolution.

[2] Liu et al., 2022, A convnet for the 2020s.

[3] Le et al, 2015, Tiny imagenet visual recognition challenge.

[4] Map of Camille Isben's contributions to Biodiversity of Magnuson Park, Seattle, WA, project, 2017, iNaturalist.
[5] Kaggle, 2015, Otto Group Product Classification Challenge.

[6] Blackard and Dean, 1990, Predicting forest cover types.

[7] Kohavi and Sahami, 1996, Error-based and entropy-based discretization.

[8] Gorishniy et al., 2021, Revisiting deep learning models for tabular data.

[9] Sekhari et al., 2021. SGD: The role of implicit regularization, batch-size, and multiple-epochs.

---

> ### Author Response · Authors · 2023-08-19
> **Additional Experiments Regarding the Role of Batch Size**
>
> Dear Reviewers and AC’s,
>
> We express our sincere gratitude for your constructive feedback and time spent reviewing our manuscript. Your insights have been invaluable.
>
> Prompted by your feedback, especially concerning the role of batch size in regularization for imbalanced training, **we have conducted additional extensive and rigorous experiments**, on top of those we performed earlier during the review process. While we can't send the figures directly, we've included detailed tables summarizing the results and sent the corresponding figures to the AC.
>
> **1. Train/Test Error Analysis for Minority and Majority Classes:**
> We present the training and testing error as a function of batch size for both minority and majority classes for Tiny-ImageNet on ConvNetXt. As depicted in the tables, during training, we observe that the minority classes tend to overfit significantly more when using bigger batch sizes. This overfitting is noticeably reduced as the batch size increases, indicating that bigger batch sizes can lead to increased overfitting, particularly for minority classes.
>
> | Batch Size | Train Error  (Minority Classes) | Train Error (Majority Classes) | Test Error (Minority Classes) | Test Error (Majority Classes) |
> |------------|--------------------------------|--------------------------------|-------------------------------|-------------------------------|
> |16| 5.7  | 4.1  | 81.7 | 36.7 |
> | 32| 5.3  | 3.3   | 81.7 | 35.8 |
> | 64| 3.1| 2.7  | 81.8   | 35.6|
> | 128 | 1.7| 2.2  | 81.9  | 35.2  |
> | 256 | 0.9 |1.9| 82.0 | 34.6   |
> | 512| 0.4  | 1.6   | 82.2 | 34.6  |
> | 1024| 0.2   | 1.4 | 82.4 | 34.8  |
> | 2048| 0.1 | 1.3  | 82.7 | 34.9 |
> | 4096| 0.06 | 1.3  | 83.1 | 34.9 |
>
> **2. Variance of Gradients Analysis for Balanced vs. Imbalanced Training:**
> The variance of neural network parameter gradients serves as a crucial indicator of the network's learning dynamics, reflecting the stability and noise in the parameter updates during training. This variance reveals a trade-off between generalization and overfitting; specifically, lower gradient variance often coincides with overfitting, while higher variance can introduce a form of implicit regularization, as is a hypothesized benefit of SGD, promoting better generalization at the cost of noisier updates.
>
> For this experiment, we investigate the variance of the gradients as a function of the batch size for Tiny-ImageNet on ConvNetXt, comparing two distinct training scenarios: balanced and imbalanced training.
> The table demonstrates that imbalanced training exhibits much lower gradient noise, especially as batch sizes grow larger. This phenomenon highlights why models are more prone to overfitting under imbalanced training, as the lower gradient noise enables the model to overfit more tightly to the training data without regularization.
>
> | Training Type      |16|32| 64|128| 256|512|1024|2048|4096|
> |--------------------|------|------|------|------|------|------|------|------|------|
> | Balanced Training|8e-2| 7e-2 |4e-2 | 9e-3 | 7e-3 | 5e-3|1e-3|8e-4 |6e-4 |
> | Imbalanced Training| 4e-2 | 2e-2 | 9e-3 | 8e-4 | 7e-5 | 3e-5 |9e-6| 3e-6 | 3e-7 |
>
> **3. Hessian Spectrum and Robustness:**
>
> Another indicator of generalization and overfitting is loss flatness, for example measured by Hessian singular values. Recent work [Yao et al. 2018b] uses the Hessian spectrum and shows that large batch  leads to convergence to points with high curvature. They characterize curvature via the dominant eigenvalue of the Hessian and suggest that high curvature can lead to poor generalization. In order to investigate it in the context of imbalanced training, we train models on balanced and imbalanced datasets for different batch sizes and calculate the top eigenvalue of the Hessian with respect to the model parameters. In the table below, we see that the top eigenvalue for imbalanced training increases much faster as we increase the batch size compared to balanced training. This suggests that higher batch sizes increase the risk of overfitting, and this phenomenon is worse for imbalanced data.
>
> | Training Type/Batch Size|16| 32|64|128|256|512|1024|2048
> |--------------------|------|------|------|------|------|------|------|------
> | Balanced Training|12|21| 36|112|173|291|404|571|
> | Imbalanced Training|15|37| 68|159|253|396|739|1428|
>
> These additional experiments offer concrete justification for the observations in our paper.  We are committed to integrating these new results into the camera-ready version, presenting them clearly through figures, and refining our discussions. We are confident that this additional analysis significantly strengthens our work, offering robust evidence and clearer insights into batch size effects in imbalanced training.
>
> We kindly ask you to consider these revisions and additional experiments in your final assessment.  Thank you again for your constructive and insightful feedback. We appreciate your consideration and welcome further suggestions.

---

### Decision · Program_Chairs · 2023-09-21

**Decision:**

Accept (poster)

**Comment:**

This paper initially received diverse scores. After the rebuttal period, most of the reviewers acknowledged their concerns addressed and raised their scores to suggest acceptance. One reviewer concerning about the limited novelty and the architecture variants tested did not respond to the authors’ rebuttal. The AC inspected the paper and all the comments and believed that these issues have been addressed by the authors. The AC agreed with the other reviewers on the quality and significant contributions of this paper, and also recommended accept.